# Robust optogenetic inhibition with red-light-sensitive anion-conducting channelrhodopsins

Johannes Oppermann[1]*, Andrey Rozenberg[2†], Thomaz Fabrin[3†],
Cristian González-Cabrera[3,4], Rafael Parker[3,4], Oded Béjà[2], Matthias Prigge[3,4,5]*,
Peter Hegemann[1]

[1]Institut für Biologie, Experimentelle Biophysik, Humboldt-Universität zu Berlin,
Berlin, Germany; [2]Faculty of Biology, Technion – Israel Institute of Technology,
Haifa, Israel; [3]Research Group Neuromodulatory Networks, Leibniz Institute for
Neurobiology, Magdeburg, Germany; [4]Aligning Science Across Parkinson's (ASAP)
Collaborative Research Network, Chevy Chase, United States; [5]Center for Behavioral
Brain Sciences, CBBS, Magdeburg, Germany

**\*For correspondence:**
johannes.oppermann@hu-berlin.
de (JO);
prigge.matthias@gmail.com (MP)

[†]These authors contributed
equally to this work

**Competing interest:** The authors
declare that no competing
interests exist.

Reviewing Editor: Jun Ding,
Stanford University, United
States

**Abstract** Channelrhodopsins (ChRs) are light-gated ion channels widely used to optically activate or silence selected electrogenic cells, such as individual brain neurons. Here, we describe identifying and characterizing a set of anion-conducting ChRs (ACRs) from diverse taxa and representing various branches of the ChR phylogenetic tree. The *Mantoniella squamata* ACR (MsACR1) showed high sensitivity to yellow-green light ($\lambda_{max}$ at 555 nm) and was further engineered for optogenetic applications. A single amino-acid substitution that mimicked red-light-sensitive rhodopsins like Chrimson shifted the photosensitivity 20 nm toward red light and accelerated photocurrent kinetics. Hence, it was named red and accelerated ACR, raACR. Both wild-type and mutant are capable optical silencers at low light intensities in mouse neurons *in vitro* and *in vivo*, while raACR offers a higher temporal resolution.

## eLife assessment

This **important** study describes the discovery and further engineering of a red light-activated, chloride-conducting Channelrhodopsin (ACR) that could be used to inhibit neuronal activity. The evidence for the spectral confirmation and biophysical characterization of MsACR and raACR, and ion selectivity are **solid**; however, the evidence supporting the use of the tools in vivo is **incomplete** and missing proper controls. In addition, benchmarking against other inhibitory tools is somewhat missing. With the in vivo part strengthened, this paper would interest neuroscientists seeking more efficient ways to inhibit neuronal activity.

## Introduction

Channelrhodopsins (ChRs) are light-activated ion channels (***Nagel et al., 2003***; ***Nagel et al., 2002***). Many of them are found in algae with flagella and eyespot (***Rozenberg et al., 2020***), where they are used for phototactic movement (***Baidukova et al., 2022***; ***Govorunova et al., 2004***; ***Sineshchekov et al., 2005***). In optogenetics, cation-conducting ChRs (CCRs) are repurposed to trigger electrical activity in excitable tissue through optically evoked membrane depolarization. To shunt electrical activity in neurons, natural and engineered anion-conducting ChRs (ACRs) are predominantly used to hyperpolarize the membrane potential in neurons (***Emiliani et al., 2022***). Most recently, potassium-selective

CCRs (KCRs) have been discovered in stramenopiles and other heterotrophic flagellates, offering additional potential to silence electrogenic cells (*Govorunova et al., 2022*; *Vierock et al., 2022*).

A limiting factor to efficiently control behavior in model organisms is the need for a large spread of light that sufficiently covers a fraction of behavior-promoting neuronal circuits. An activation wavelength ($\lambda_{max}$) in the red spectral range allows penetration in deeper layers than blue light (*Ash et al., 2017*). Therefore, optogenetic tools that are sensitive to red light are desirable. Currently, RubyACRs from the Labyrinthulea are the most redshifted ACRs reported. However, they exhibit a substantial photocurrent inactivation and a slow photocurrent recovery at high light intensities (*Govorunova et al., 2020*) caused by the accumulation of long-lasting and non-conducting secondary photointermediates (*Sineshchekov et al., 2023*). Furthermore, the operational light sensitivity that describes the ability of an optogenetic tool to exert an effect on the cell at a given photon density also defines the volume of activated neuronal tissue. Here, ACRs that combine high expression, high single-channel conductance, and moderate kinetics, resulting in a high operational light sensitivity, are ideal for efficiently silencing larger volumes.

We screened a panel of ChRs across the phylogenetic tree, including previously unexplored ChR clades. Among them, MsACR1, a member of the prasinophyte ACR family (*Rozenberg et al., 2020*), was the most activated with yellow-green light of 555 nm. Introducing substitutions into the wild-type MsACR1 protein revealed a single mutation that caused a further redshift of the action spectrum and accelerated the photocurrent kinetics. Furthermore, we evaluated the optogenetic usability of MsACR1 and its mutant raACR for 'red and accelerated ACR'. As such, we recorded dissociated hippocampal neurons at various light stimulation paradigms and tested the possibility of disrupting neuronal network activity in the primary motor cortex that directly influences locomotion behavior. The ACRs are well expressed and reliably suppress activity with nano-watt light intensities. Additionally, raACR offers suppression of action potentials at a higher temporal resolution than MsACR1.

## Results

### Characterization of metagenomically identified ACRs

The expressed opsin candidates represent multiple ChR subfamilies, some coming from previously unexplored clades: MsACR1 from the mamiellophycean green alga *Mantoniella squamata*, a member of the prasinophyte ACR family (*Rozenberg et al., 2020*); GpACR1, a protein from the heterotrophic cryptomonad *Goniomonas pacifica* unrelated to cryptophyte ACRs and instead putatively related to labyrinthulean ACRs; S16-ACR1, which we discovered in the single-amplified genome (SAG) of the uncultured MAST3 stramenopile S16 (*Wideman et al., 2020*), representing its own deeply branching clade; TaraACR3, a haptophyte ACR of metagenomic origin; and TaraACR1 and TaraACR2, two related ChR sequences coming from giant viruses (see Supplementary Discussion) and forming a separate ChR subfamily (*Figure 1A*, and *Appendix 1—figure 1*).

We expressed the opsin domains of the identified putative ChRs in ND7/23 cells to test for electrical activity by voltage-clamp electrophysiology in whole-cell mode. We determined the wavelength sensitivity, photocurrent amplitudes, kinetics, and ion selectivity. All constructs showed bidirectional photocurrents upon illumination, typical for ChRs. However, while MsACR1, TaraACR1, and TaraACR2 showed no or only little photocurrent reduction under continuous illumination (photocurrent inactivation), TaraACR3, S16-ACR1, and GpACR1 showed considerable inactivation. In particular, S16-ACR1 photocurrents showed almost complete inactivation similar to the previously described MerMAIDs and some cryptophyte CCRs (*Oppermann et al., 2019*; *Sineshchekov et al., 2020*). This effect was more pronounced at positive holding potentials. GpACR1 photocurrents, on the other hand, elicited a secondary peak photocurrent upon light-off (*Figure 1D*) indicative of photointermediate inactivation during illumination (*Berndt et al., 2009*; *Krause et al., 2017*).

Most of the tested ChRs showed a $\lambda_{max}$ between 470 and 510 nm, while MsACR1 exhibited a considerable redshifted $\lambda_{max}$ at 555 nm (*Figure 1B, C*, and *Appendix 1—figure 3A*). At –60 mV holding potential, photocurrent amplitudes of up to 10 nA were elicited (*Figure 1E*), decaying with an off-kinetic ($\tau_{off}$) between 3 ms (TaraACR2) and 400 ms (GpACR1) (*Figure 1F*). The operative light sensitivity was proportional to $\tau_{off}$. TaraACR3, GpACR1, and MsACR1, with slow off-kinetics, had a high light sensitivity with half-maximal activation at 20–40 µW/mm². In contrast, TaraACR2 and S16-ACR1

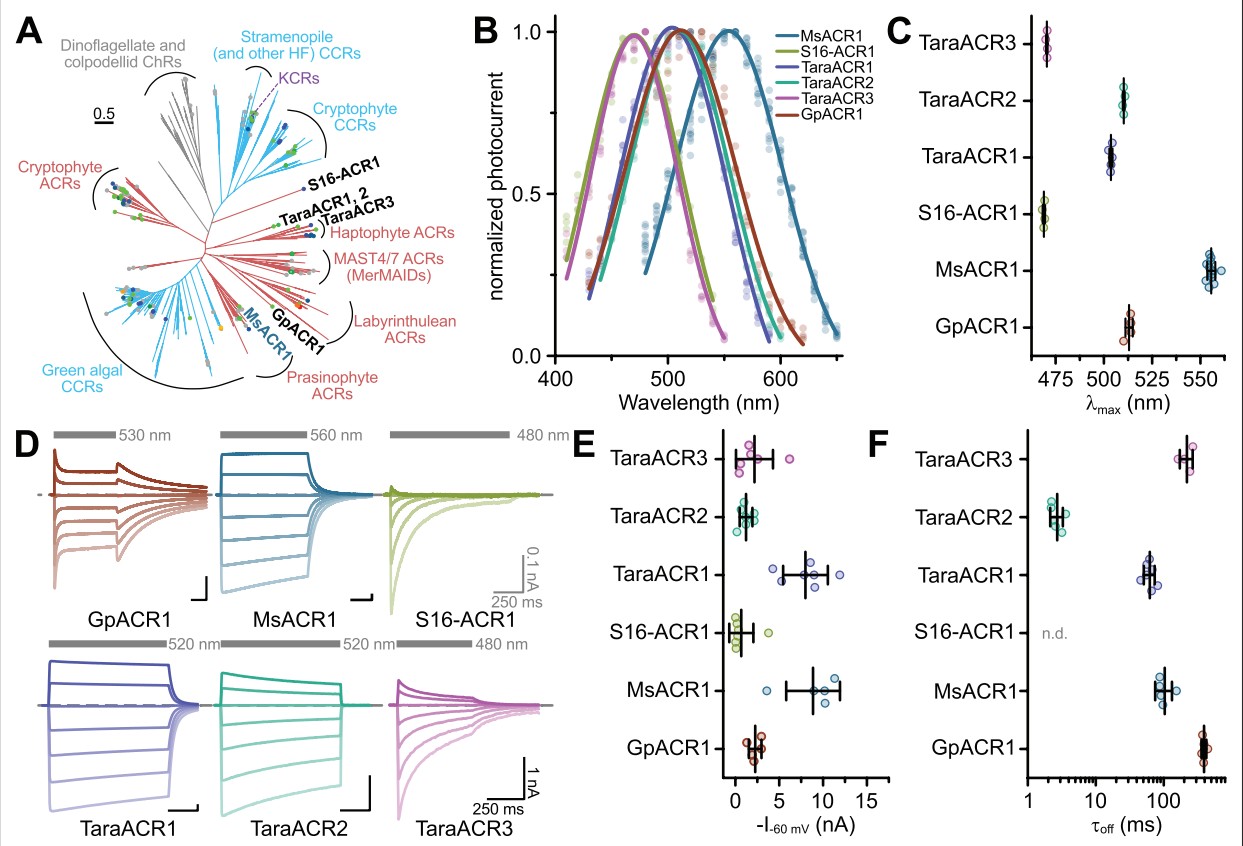

**Figure 1.** Phylogeny and electrophysiological characterization of identified anion-conducting channelrhodopsins (ACRs). (**A**) Phylogenetic relationships among known channelrhodopsins (ChRs) subfamilies with the described ChRs written in black and blue font. The color of the clades indicates ion selectivity: red for ACRs and light-blue for cation-conducting ChRs (CCRs). The color of the dots reflects $\lambda_{max}$ with gray dots corresponding to proteins not yet having their $\lambda_{max}$ determined. See the extended version of the tree in *Appendix 1—figure 1*. (**B**) Action spectra of identified ACRs. Shown are Gaussian fits, where dots are data points of single measurements. (**C**) Strongest activation wavelength ($\lambda_{max}$) derived from Gaussian fits of recorded action spectra. (**D**) Example current traces of identified ACRs recorded at holding potentials between −80 and +40 mV. The gray bars indicate the illumination period with excitation light of indicated wavelength. The dashed line indicates 0 nA. (**E**) Peak photocurrent amplitude at −60 mV holding potential. (**F**) Off-kinetics ($\tau_{off}$) of the photocurrents upon light-off at −60 mV for tested ACR candidates. Note: $\tau_{off}$ for S16-ACR1 could not be determined because photocurrents fully inactivated at −60 mV during the illumination. In (**C**, **E**, **F**), black lines are mean ± standard deviation, and circles are data points of single measurements (GpACR1 $n$ = 4, 5, 5; MsACR1 $n$ = 12, 5, 5; S16-ACR1 $n$ = 4, 7, n.d.; TaraACR1 $n$ = 5, 7, 7; TaraACR2 $n$ = 4, 7, 6; TaraACR3 $n$ = 4, 6, 4 for (**C**), (**E**), and (**F**), respectively). Abbreviation: n.d. means not determined.

The online version of this article includes the following source data for figure 1:

**Source data 1.** Source data for electrophysiological analysis of identified anion-conducting channelrhodopsins presented in *Figure 1B, C, E, F*.

photocurrents did not saturate at light intensities up to 4 mW/mm², demonstrating a low operative light sensitivity (*Appendix 1—figure 3B, C*).

To determine the ion selectivity of the ChRs, either the external Cl⁻ ([Cl⁻]$_{ex}$) was replaced with Asp⁻ or the external Na⁺ ([Na⁺]$_{ex}$) with NMDG⁺. Only the replacement of [Cl⁻]$_{ex}$, but not [Na⁺]$_{ex}$, resulted in photocurrent changes and a substantial shift in the reversal potential ($E_{rev}$) close to the Nernst potential ($E_{Nernst}$) of Cl⁻ (*Appendix 1—figure 3D–F*). Therefore, all tested ChRs are ACRs.

## Single-residue mutations affect the spectral sensitivity of MsACR1

The strong activation of MsACR1 with yellow-green light prompted us to explore its applicability as an optogenetic silencer. Therefore, we tested several point mutations to shift its sensitivity to the red. Generally, a redshift can be achieved by either increasing the polarity close to the β-ionone ring of the retinal chromophore or reducing the polarity in the proximity of the retinal Schiff base (RSB).

First, we tested mutations in the vicinity of the β-ionone ring (*Figure 2A*). The exchange of the conserved Proline in this region (HsBR-P186; see *Appendix 1—figure 2* for an alignment) generally

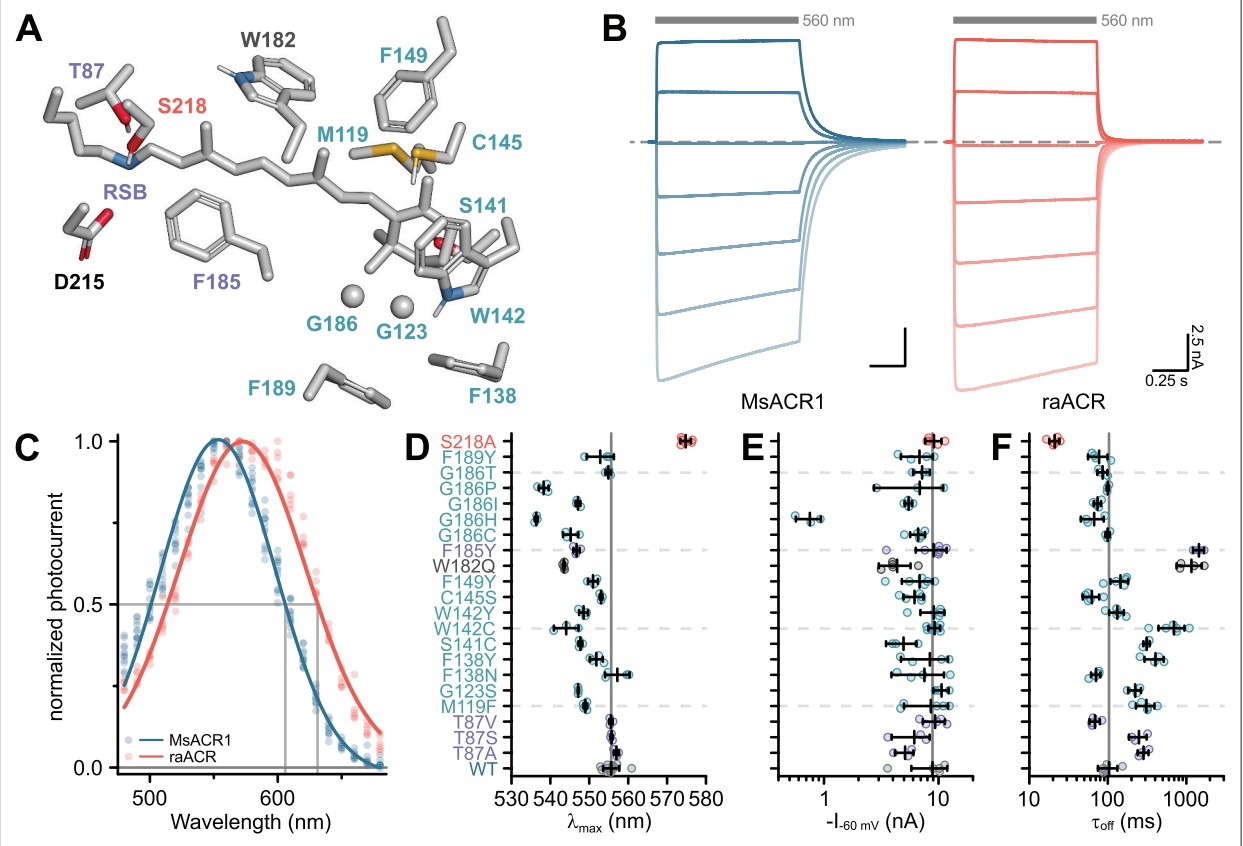

**Figure 2.** Color-tuning of MsACR1. (**A**) Structural model of the MsACR1 retinal binding pocket. Residues close to the β-ionone ring are labeled in cyan, and those close to the retinal Schiff base (RSB) are labeled purple. Residue S218, the position of the raACR mutation, is labeled in light-red, and the putative counterion D215 is labeled black. Residue side chains and the all-trans-retinal chromophore are shown as sticks except for G123 and G186, where the Cα is shown as a sphere. Oxygen is colored in red, nitrogen in blue, and sulfur in yellow. (**B**) Example current traces of MsACR1 (blue) and raACR (MsACR1-S218A; light-red). Gray bars above the traces indicate the illumination period with an excitation light of 560 nm. The dashed line indicates 0 nA. Example current traces of all mutants can be found in *Appendix 1—figure 4*. (**C**) Action spectra of MsACR1 (blue) and raACR (red). Dots are data points of single measurements fitted with a Gaussian function (solid line). (**D**) Strongest activation wavelength ($\lambda_{max}$) of the indicated single-residue mutants. (**E**) Photocurrent amplitudes at −60 mV holding potential ($I_{-60\,mV}$) of the indicated mutants upon excitation with 560 nm light. (**F**) Off-kinetics ($\tau_{off}$) of the photocurrents upon light-off. In (**D–F**), black lines are mean ± standard deviation, and circles are data points of single measurements (WT $n$ = 12, 5, and 5, respectively; Mutants $n$ = 3–7).

The online version of this article includes the following source data for figure 2:

**Source data 1.** Source data for electrophysiological analysis of color-tuning in MsACR1 presented in *Figure 2C-F*.

causes strong blue-shifts (*Lu et al., 2001*). However, the converse G186P mutation in MsACR1 also blue-shifts the activity (*Figure 2D*), as do all other tested replacements (G186C, G186H, and G186T), with a substantial amplitude reduction in the case of G186H (*Figure 2E*). Other tested replacements close to the β-ionone ring either did not affect $\lambda_{max}$ or resulted in hypsochromic shifts, including mutations mirroring residues unique to the recently described RubyACRs: W142Y, W182Q, and G186I (*Figure 2D*, *Govorunova et al., 2020*). The only exception was F138N, which showed a small redshift (*Figure 2D*) but also strongly increased $\tau_{off}$ (*Figure 2F* and *Appendix 1—figure 4*).

Next, we focused on the active site close to the RSB. Replacing F185, a residue oriented toward the putative counterion D215 (*Figure 2A*), with a Tyrosine (F185Y) also resulted in a blue-shifted $\lambda_{max}$ (*Figure 2D*) and increased $\tau_{off}$ (*Figure 2F* and *Appendix 1—figure 4*). Substitution of T87 with Alanine led to a minor redshift (*Figure 2D*) but decreased photocurrent amplitudes and decelerated $\tau_{off}$ (*Figure 2E, F* and *Appendix 1—figure 4*, *Oda et al., 2018*). Introduction of Valine or Serine (T87V and T87S) did not affect the spectral sensitivity (*Figure 2D*).

Previously, in Chrimson, the residue A298 next to the retinal-binding Lysine was associated with the redshifted absorption of this ChR (*Oda et al., 2018*). In MsACR1, the homologous residue is S218.

Replacing it with an Alanine (MsACR1-S218A) reduced $\tau_{off}$ fivefold (*Figure 2F* and *Appendix 1—figure 4*) and shifted $\lambda_{max}$ 20 nm toward longer wavelengths (*Figure 2D* and *Appendix 1—figure 5A*). Consequently, the half-maximal activity on the red side of the spectrum also shifted from 605 to 635 nm (*Figure 2C*). Additionally, this mutation increased the photocurrent amplitude at 690 nm twofold in comparison with the wild-type. Relative to GtACR1, the most commonly used ACR for optogenetic application (*Mahn et al., 2016*; *Tsunoda et al., 2017*), MsACR1-S218A exhibits a 19-fold larger photocurrent at 690 nm (*Appendix 1—figure 5C*). Due to these advantageous properties, we coined the mutant raACR for '<u>r</u>ed and <u>a</u>ccelerated <u>ACR</u>'.

## raACR allows efficient shunting in neurons at nano-watt light powers

Next, we explored neuronal applications of both MsACR1 and raACR in an *in vitro* neuronal preparation. We then transduced dissociated hippocampal neurons with adeno-associated viruses carrying the ACRs under the control of a pan-neuronal human Synapsin I (hSyn1) promoter. After 10 days, membrane-localized fluorescence was visible in somatic, axonal, and dendritic compartments without apparent intracellular or vesicular aggregation (*Figure 3A*).

Current-evoked spiking was strongly suppressed in single neurons by 550 nm light at intensities of >200 nW/mm$^2$. Excitation of MsACR1 at just 5 nW/mm$^2$ light already decreased the spiking probability by 25%. At light intensities between 100 and 7100 nW/mm$^2$, evoked activity was further reduced to 30–90% for both ACRs (*Figure 3B, C* and *Appendix 1—figure 6*). To quantify the shunting efficiency at different illumination paradigms, we either applied 550 or 635 nm light continuously or pulsed at 10 Hz (2 ms pulse width) during a current-ramp injection (*Figure 3D*; *Tsunoda et al., 2017*). Notably, the pulsed illumination reduces the photon exposure for the neurons 50-fold compared to continuous illumination. MsACR1 and raACR both completely silenced current-evoked spiking during continuous illumination (*Figure 3E*). However, at the on-set of the light pulse, we observed a light-evoked on-set spike (*Figure 3D*). This has been similarly described for other ACRs and arises in axonal compartments with a higher intracellular chloride concentration compared to the soma, triggering initial chloride efflux upon ACR activation that leads to a short depolarization (*Mahn et al., 2016*).

To avoid misinterpretations arising from an on-set spike during *in vivo* optogenetic experiments, we designed and tested soma-targeted (st) versions of MsACR1 and raACR. For this, we fused a short targeting sequence from the voltage-gated potassium channel Kv2.1 to the N-terminus of the ACRs. We determined the on-set spike probability for st-MsACR1 and st-raACR under continuous and pulsed illumination with 550 nm. The latter significantly reduced the on-set spike probability for both variants (*Appendix 1—figure 7C*). Furthermore, compared to their untargeted versions, st-raACR showed a significantly reduced on-set spike probability, while st-MsACR1 showed a trend of a reduced on-set spike probability (*Appendix 1—figure 7D*). MsACR1 and st-MsACR1 also nearly completely shunt all spikes during pulsed illumination independent of the light color. Activation of raACR, or its soma-targeted version, significantly reduced the number of spikes under all conditions but with a lower potency compared to MsACR1 (*Figure 3E* and *Appendix 1—figure 7B*), likely due to the reduced ion flux during a single photocycle.

We further assessed the minimum duration required to re-evoke a current-induced spike after optically suppressing a preceding action potential (*Figure 3F*). raACR allows spike triggering with a 50% spiking frequency already 50 ms after optical silencing, whereas upon activation of MsACR1, a 50% spiking frequency was reached only 180 ms post-shunting (*Figure 3G*).

To assess the potential use of MsACR1 and raACR *in vivo*, we first modeled the brain tissue volume that could be effectively silenced. We used a Monte Carlo simulation and our *in vitro* light titration data (*Stujenske et al., 2015*). The comparison with other commonly used optogenetic attenuators (Jaws [*Chuong et al., 2014*] and GtACR1 [*Govorunova et al., 2015*]) showed the superiority of MsACR1 and raACR (*Appendix 1—figure 8A, B*). To evaluate the efficiency of MsACR1 and raACR *in vivo*, we assessed how light-induced transient neuronal network perturbation in the primary motor cortex (M1) translates into locomotion impairments. As such, we injected high-titer viral particles of the ACRs into the M1 of wild-type C57BL/6 mice and implanted a 200-µm optical fiber above the injection site (*Figure 4A*).

Both ACRs exhibit homogenous expression in cortical neurons without visible intracellular aggregates or cell death (*Figure 4A*). Two-second illumination bouts with 635 nm light pulsed at 20 Hz led to immediate locomotion arrest in all tested animals (*Figure 4B, C*). We quantified these motor

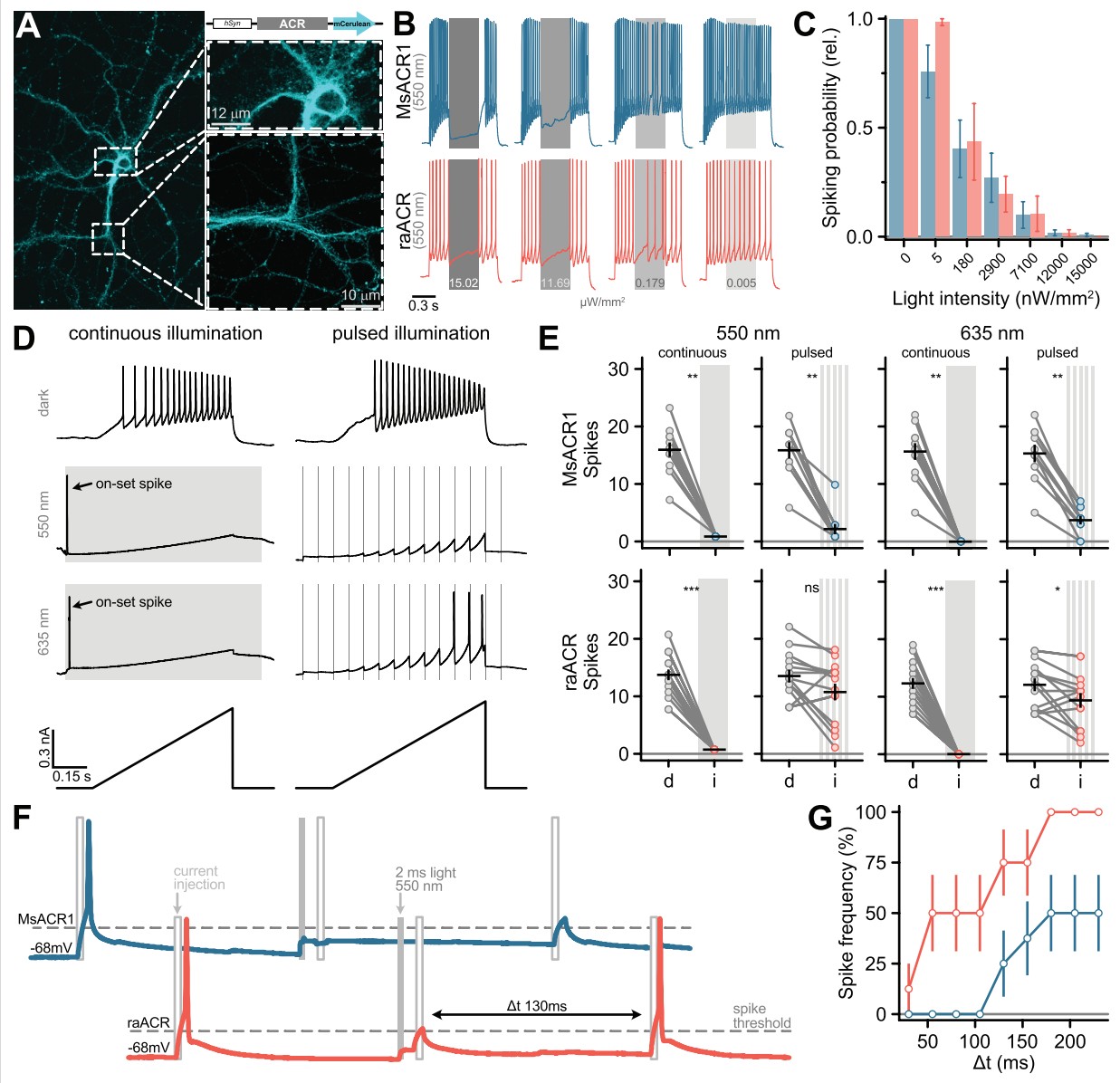

**Figure 3.** Silencing of neuronal activity in dissociated hippocampal neurons using MsACR1 and raACR. (**A**) Confocal image of a hippocampal neuron 10 days after viral transduction expressing MsACR1-mCerulean under control of the hSynapsin1 (hSyn) promoter. The magnified portion of the image of a putative axonal compartment (right side, lower panel) shows a high degree of membrane targeting of MsACR1. Magnification of the somatodendritic compartment (right side, upper panel) reveals the absence of visible protein aggregation. (**B**) Voltage traces of neuronal spiking elicited by a 1-s 250-pA current step illustrating the efficiency of optical silencing at indicated various light intensities (ranging from 200 to $15 \times 10^3$ nW/mm²). MsACR1 and raACR were activated for 300 ms with 550 nm light (gray bars) during the current step. (**C**) Relative neuronal spiking probability upon optical silencing by MsACR1 (blue) or raACR (red) excited with 550 nm light, as illustrated in (**B**), at all tested light intensities. (**D**) Voltage traces of MsACR1-expressing hippocampal neurons upon current ramps under no light, continuous or 10 Hz pulsed illumination with 550 or 635 nm light. Continuous illumination evoked a spike at light on-set (on-set spike). (**E**) Spikes elicited by current-ramp injection in the dark (d) or upon continuous or pulsed illumination (i) of 550 or 635 nm in neurons expressing MsACR1 (upper row) or raACR (lower row), as shown in (**D**). Black lines are mean ± standard error, and circles represent single measurements (*n* = 10–16). Paired two-tailed Wilcoxon or Student's *t*-tests were used for MsACR1 and raACR comparisons (ns: p > 0.05, *: p <= 0.05 **: p <= 0.01 ***: p <= 0.001). (**F**) Temporal precision of neuronal inhibition of MsACR1 (blue) and raACR (red). A paradigm was designed to measure the period after successful suppression of a current-evoked spike in which no new spike can be triggered. (**G**) Population data of MsACR1 (blue) and raACR (red) for the experiment are shown in (**F**).

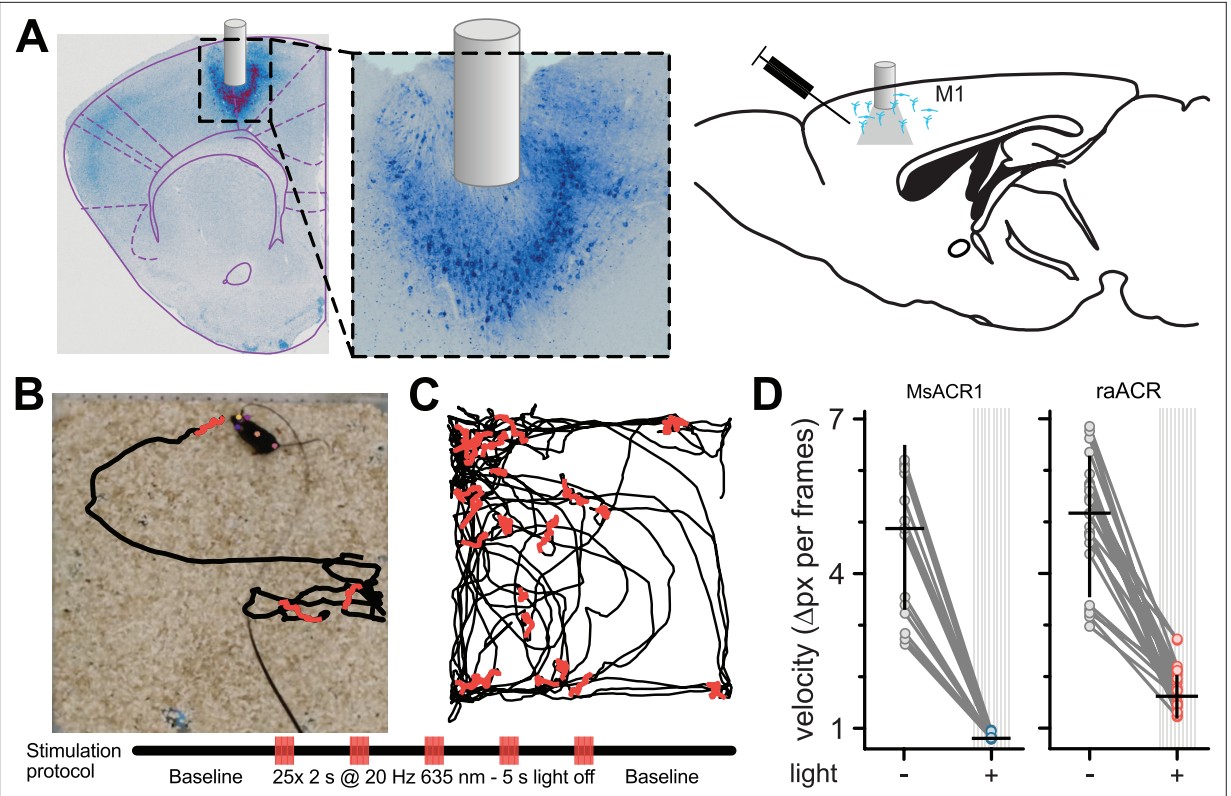

**Figure 4.** Inhibition of locomotion in rodents *in vivo* using MsACR1 and raACR. (**A**) Confocal images of coronal brain slices confirm the expression of raACR in the primary motor cortex (M1) and the accurate placement of the optical cannula. A close-up of the injection site confirms homogenous expression in the transduced brain volume with no apparent neuronal degeneration. (**B**) Experimental design and stimulation protocol. Animals were allowed to explore the arena for 1 min before stimulation at 20 Hz with 635 nm for 2 s. (**C**) The trajectory of a mouse for bouts with no illumination (black) and periods during light stimulation (red). (**D**) Calculated velocity based on pixel deviation during two consecutive frames for periods with illumination (+) and no illumination (Mann–Whitney *U* = 23, p < 0.01).

arrests by calculating the pixel movement of a body marker between two consecutive frames during periods without and with illumination (*Figure 4D*). Activation of either ACR led to significantly reduced velocity. However, the effect was more substantial for MsACR1, likely due to its longer-lasting response compared to raACR (*Figure 3G*).

As complete locomotion arrest upon light stimulation points to a complex disruption of M1 circuit function, we decided to dissect this complexity by targeting expression exclusively to parvalbumin-positive inhibitory neurons (PV-Cre) utilizing our soma-targeted versions. Additionally, we implanted a 32-channel single-wire optrode to measure neuronal network responses *in vivo* upon illumination directly (*Anikeeva et al., 2012*; *Figure 5A*). Six weeks post-surgery, we measured neuronal responses in the M1 area during a 2-s light pulse while the animals were freely moving in an open field arena. As expected, we observed complex changes in neuronal firing, ranging from an immediate decrease or increase in neuronal spiking, delayed changes in firing rate during light stimulation, and resetting of the firing rate after stimulation (*Figure 5B*). To capture the different response modes, we calculated the *Z*-score for 100 ms binned firing rates for all recorded units during the 2 s before, during, and after illumination with blue (473 nm) and red light (645 nm) at both 1 and 10 mW (*Figure 5C, D*). During the first 100 ms within a blue light stimulation at 1 mW, 9 out of 33 units for MsACR1 showed at least a twofold change in standard deviation (SD) compared to baseline (2 s before illumination), which increased to 16 units at 10 mW. For illumination with red light at either 1 or 10 mW, more than 75% of all units (51 units) responded with a change in firing rate. Interestingly, the fast-cycling raACR shows a superior inhibition at blue and red illumination regimes at 10 mW compared to MsACR1, exhibiting a modulation of neuronal spiking in more than 90% of units. At light intensities of 1 mW for both wavelengths, raACR showed moderate effects on the cortical activity of 35%. In both cases, for MsACR1 and raACR, red light illumination leads to a significant decrease in running speed at 10 mW

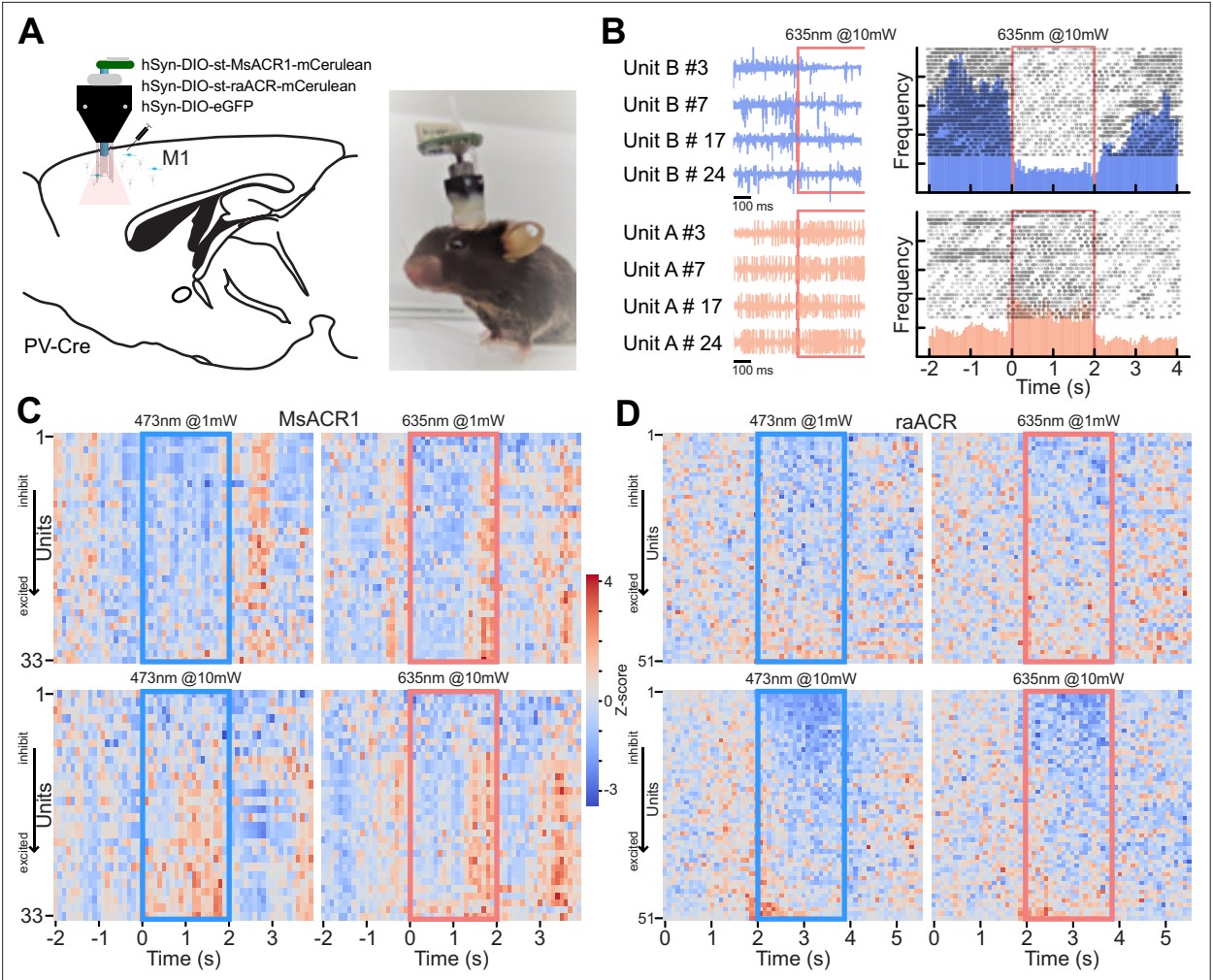

**Figure 5.** Activation of st-MsACR1 and st-raACR in PV-Cre neurons in M1 leads to complex changes in firing rate. (**A**) Illustration of the injection strategy, viral constructs, optrode design, and implantation in PV-Cre mice. The diagram shows the targeting of M1 with hSyn-DIO-st-MsACR1-mCerulean, hSyn-DIO-st-raACR-mCerulean, and hSyn-DIO-eGFP. (**B**) Recordings from a single electrode 500 ms before and during light stimulation of raACR at 10 mW, 635 nm, in four repeats. Blue traces (Unit B) exhibit a decrease in spiking rate compared to baseline firing during illumination, while traces from Unit A show a repeated increase in firing rate. The peri-stimulus time histogram (PSTH) confirms clear inhibition during light stimulation over the entire 2-s period (upper, blue histogram) and an increase in firing rate (lower, red histogram). (**C, D**) Summary of the entire dataset as ranked Z-scores for 33 units expressing MsACR1 and 51 units expressing raACR, respectively. Units that respond with reduced spiking upon illumination are displayed at the top of the heat map, while units that respond with increased spiking are shown at the bottom. The heat maps show the Z-scored firing rates for blue light (473 nm) at 1 and 10 mW and red light (635 nm) at 1 and 10 mW.

light intensities. Control animals injected with hSyn1-DIO-eGFP do not show a significant change (Mann–Whitney $U$, p > 0.05 control, p < 0.001 for MsACR1 and raACR) (*Appendix 1—figure 9A, B*).

## Discussion

Despite the rapidly evolving optogenetic toolbox that offers versatile possibilities to manipulate the electrical activity of excitable cells, highly efficient red-light-activatable tools are still sparse. We identified MsACR1 and tailored a red-light-sensitive ACR with large photocurrent amplitudes and fast off-kinetics that we coined raACR. We further demonstrated the applicability and potency of both wild-type and mutant in several optogenetic experiments.

Besides its availability as a useful optogenetic tool, the wild-type MsACR1 may also offer a venue for answering questions about algal phototaxis. Previously, we suggested that in motile prasinophytes, both ACRs and CCRs may cooperate to regulate photo-orientation (*Rozenberg et al., 2020*).

Recently, proton conduction has been shown for CCR1 from *M. squamata* (*Govorunova et al., 2021*). Therefore, *M. squamata* may be a suitable algae species to investigate the physiological function of proton- and chloride-carried photocurrents.

Furthermore, among the characterized ACRs are two of viral origin, TaraACR1 and TaraACR2. These proteins constitute a further contribution to the growing number of rhodopsins from viruses (*Yutin and Koonin, 2012*; *Bratanov et al., 2019*; *Needham et al., 2019*; *Rozenberg et al., 2020*; *Zabelskii et al., 2020*; *Hososhima et al., 2022*). Compared to some of the previously characterized viral rhodopsin channels (*Rozenberg et al., 2020*; *Zabelskii et al., 2020*), the two described here were well expressed and showed large photocurrents. Also, TaraACR2 showed fast photocurrent off-kinetics comparable to ZipACR and RapACR (*Govorunova et al., 2018*; *Govorunova et al., 2017*).

While GpACR1 and S16-ACR1 are unsuitable for optogenetic applications due to their photocurrent amplitudes and kinetics, their photocurrents unravel interesting molecular features. GpACR1 photocurrents elicit a second peak upon light shut-off, a behavior seen before for the slow-cycling CrChR2 variant C128S (*Berndt et al., 2009*), ReaChR (*Krause et al., 2017*), or the viral rhodopsin VirChR1 (*Zabelskii et al., 2020*). Its appearance is expected to be caused by the back photoisomerization of photocycle intermediates (*Krause et al., 2017*). S16-ACR1, on the other hand, shows a substantial voltage-dependent photocurrent inactivation and nearly complete inward rectification similar to MerMAIDs (*Oppermann et al., 2019*) and may offer further insight into these processes in general.

The action spectrum of MsACR1 shows its highest activity in yellow-green light and a distinct blue-light peak, probably resembling the β-band absorption of the retinal chromophore originating from two-photon absorption (*Stavenga et al., 1993*). Surprisingly, most of the substitutions in MsACR1 close to the β-ionone ring resulted in a blue-shift. Only one mutation (F138N) caused a small redshift, suggesting that this region is already relatively polar. Alternatively, the β-ionone ring could be twisted, which decouples it from the rest of the polyene chain (*Kato et al., 2015*). In contrast, the active site around the RSB was more susceptible to modification, although the effect of MsACR1-T87A was less pronounced than for Chrimson (ChrimsonSA) (*Oda et al., 2018*). However, upon replacing S218 with Alanine, the action spectrum was shifted by 20 nm toward longer wavelengths, mimicking several orange- or red-light-absorbing ChRs (*Oda et al., 2018*). Similar shifts have been achieved naturally and artificially in animal rhodopsins (*Liénard et al., 2021*).

Both MsACR1 and raACR reliably shunt spiking *in vitro* and induced motor arrest *in vivo* using 635 nm light. Shunting during continuous illumination was complete, though light-evoked on-set spikes observed for both ACRs were minimized with pulsed illumination and soma targeting of MsACR1 and raACR. Presumably, longer photon exposure is required to evoke this spike, as previously evident for GtACR1 using 10 ms light pulses (*Mahn et al., 2016*). Therefore, shorter or ramped-on light pulses can reduce the probability of such events. Overall, pulsed illumination and soma targeting are recommended for artifact-free *in vivo* optogenetic experiments.

Pulsed illumination was more efficient for MsACR1. This was due to its slower kinetics, resulting in increased ion flux per photocycle. On the other hand, the accelerated kinetics of raACR allows for a higher temporal resolution, enabling current-evoked spiking already 50 ms post-illumination. Ideally, one would slow its kinetics to increase the efficiency of raACR under pulsed illumination. However, most mutations tested here that resulted in slowed kinetics also shifted $\lambda_{max}$ to the blue. Potential candidates for combining could be T87A or T87S, as they did not affect $\lambda_{max}$ but increased $\tau_{off}$ while the current amplitudes were only marginally reduced. On the other hand, screening mutations of the homologous residues to the DC pair of CrChR2 could be fruitful, as recently demonstrated (*Rodriguez-Rozada et al., 2022*).

Also, in contrast to other ACRs commonly used as optogenetic silencers, both MsACR1 and raACR do not show substantial photocurrent inactivation during continuous illumination and can be activated again virtually instantly without photocurrent decline. This recommends them as reliable optogenetic actuators. Furthermore, the redshifted raACR reduces potential cross-activation when used with blue-light-sensitive optogenetic actuators or reporter systems (*Vierock et al., 2021*).

In conclusion, MsACR1 and raACR, due to their good expression and membrane targeting, high chloride conductance and slow off kinetics provide a high operational light sensitivity to neuronal cells which enables inhibition at light intensities as low as 20 nW/mm². Thus, we believe both ACRs are a

valuable addition to the optogenetic toolbox when long-lasting inhibition in larger brain volumes is required.

# Methods

## Identification and phylogenetic analysis

Two of the ChR sequences characterized here have been identified in *Rozenberg et al., 2020*: MsACR1 from *M. squamata* CCAP 1965/1 (1KP_QXSZ_2010086.p1, extracted from a 1KP transcriptome assembly *One Thousand Plant Transcriptomes Initiative, 2019*) and GpACR1 from *G. pacifica* (the originally reported truncated sequence MMETSP0108_DN24423_c0_g1_i1.p1 was extended by transcriptome re-assembly of the raw data from MMETSP0108 and MMETSP0107 *Keeling et al., 2014*). S16-ACR1 was extracted from the SAG S16 assigned to the marine stramenopile clade MAST3g (PRJNA379597-S16_NODE_311_gmes_4206, the coding sequence was predicted with GeneMarkS in the SAG assembly from *Wideman et al., 2020*). Sequences TaraACR1 (from contig SAMEA2622336_1814029), TaraACR2 (contig SAMEA2619952_516956), and TaraACR3 (overlapping contigs SAMEA2621277_718384 and SAMEA2621278_296705) were retrieved from a custom assembly of the *Tara* Oceans data (*Philosof et al., 2017*).

For phylogenetic reconstruction, rhodopsin domain sequences of all records included in the Catalog of Natural Channelrhodopsins v. 1.3 (*Rozenberg, 2023*), covering characterized and uncharacterized proteins, were taken and duplicated domains were excluded. The domain sequences were aligned with mafft v. 7.520 (automatic mode) (*Katoh et al., 2002*), the alignment was trimmed with trimal v. 1.4.1 (-gt 0.1) (*Capella-Gutiérrez et al., 2009*), and the phylogeny was reconstructed with iqtree2 v. 2.2.0.3 (-pers 0.2 -nstop 500) with 1000 ultrafast bootstrap replicates (*Hoang et al., 2018*; *Minh et al., 2020*).

Structure-supported alignment of the rhodopsin domains for the tested ChRs and reference proteins (BR, ChR2, Chrimson, and GtACR1) was performed with t_coffee v. 13.45.0.4846264 (algorithms sap_pair, mustang_pair, t_coffee_msa, and probcons_ms). For the reference proteins available, crystal structures were utilized, while for the ChRs characterized here, structural models were obtained with AlphaFold (*Jumper et al., 2021*) as implemented in the colabfold workflow (*Mirdita et al., 2022*). Transmembrane regions of the α-helices of ChR2 (PDB: 6EID) were taken from the OPM database (*Lomize et al., 2012*).

The bioinformatics workflow is available from the GitHub repository, https://github.com/BejaLab/ACRs (copy archived at *Rozenberg, 2024*).

## Molecular biology

For electrophysiological analysis, human/mouse codon-optimized genes coding for the identified ChRs were ordered (GenScript, Piscataway, NJ or Integrated DNA Technologies, Coralville, IA) and cloned in frame with mCherry, mScarlet, or mCerulean fluorophores into pEGFP-C1 or pcDNA3.1 vectors using Gibson cloning (*Gibson et al., 2009*) or restriction digest (FastDigest *NheI* and *AgeI*, Thermo Fisher Scientific, Waltham, MA). The membrane localization of GpACR1 and S16-ACR1 was improved using a membrane trafficking sequence and endoplasmic reticulum release sequence, as previously used (*Grimm et al., 2018*; *Rozenberg et al., 2020*). Single residues in MsACR1 were replaced by site-directed mutagenesis using *Pfu* polymerase (Agilent Technologies, Santa Clara, CA). A structural model of MsACR1 was generated using AlphaFold (*Jumper et al., 2021*), and the retinal chromophore was added from BR (PDB: 2AT9).

MsACR1 and raACR, in frame with a mCerulean and separated by a P2A site, were subcloned into pAAV2 for expression in neurons under control of the Calmodulin kinase II promoter using Gibson cloning (*Gibson et al., 2009*) and further subcloned into pAAV2 under the control of the human synapsin promoter and flanked by lox71 and lox66 sites using *EcoRI* and *BamHI* restriction sites (FastDigest, Thermo Fisher Scientific). The constructs were soma-targeted using the Kv2.1 targeting domain.

Molecular cloning was planned using NEBuilder v. 2.2.7+ (New England Biolabs Inc, Ipswich, MA) and SnapGene v. 5.0+ (GSL Biotech LLC, Chicago, IL). Plasmids of all constructs used in this study were deposited at Addgene: MsACR1-pmCerulean3-N1 (204958), raACR-pmCerulean3-N1 (204959), TaraACR1-pmCherry-C1 (204960), TaraACR2-pmCherry-C1 (204961), TaraACR3-pmCherry-C1

(204962), S16-ACR1-pcDNA3.1-mScarlet (204963), GpACR1-pcDNA3.1-mScarlet (204964), MsACR1-mCerulean3-pAAV2-hSyn (204965), raACR-mCerulean3-pAAV2-hSyn (204966), MsACR1-TS-Kv-P2A-mCerulean3-pAAV-CaMKII (204970), and raACR-TS-Kv-P2A-mCerulean3-pAAV-CaMKII (204971).

## ND7/23 cell culture

For electrophysiology, measurements of the identified ChRs were expressed in a mouse neuroblastoma and rat neuron hybrid cell line (ND7/23 cells, ECACC 92090903, RRID: CVCL_4259, Sigma-Aldrich, Munich, Germany). Cells were cultured as previously described (*Grimm et al., 2017*) at 37°C and 5% $CO_2$ in Dulbecco's modified Eagle's medium (Sigma-Aldrich) supplemented with 5% fetal bovine serum and 100 µg/ml penicillin/streptomycin (both Sigma-Aldrich). For experiments, cells were seeded at a density of $0.4 \times 10^5$ cells/ml on poly-D-lysine-coated coverslips and supplemented with 1 µM all-*trans*-retinal (Sigma-Aldrich). After 1 day, cells were transiently transfected with 2 µg DNA by lipofection (FuGENE HD, Promega, Madison, WI).

## Electrophysiology in ND7/23 cells

One to two days after transfection, photocurrents were recorded using the patch-clamp method in whole-cell mode at room temperature. The whole-cell configuration was established at ≥1 GΩ membrane resistance, and cells were discarded at access resistance >10 MΩ. Patch pipettes were pulled from borosilicate glass capillaries (G150F-8P, Warner Instruments, Hamden, CT) to resistances of 1.5–3 MΩ using a micropipette puller (P-1000, Sutter Instruments, Novato, CA) and fire-polished. During measurements, a 140-mM NaCl agar bridge was used as the reference electrode. A 2-kHz Bessel filter was used for signal amplification (AxoPatch200B). Signals were digitized at 10 kHz (DigiData400) and acquired using Clampex 10.4 software (all Molecular Devices, Sunnyvale, CA). A Polychrom V (TILL Photonics, Planegg, Germany) with 7 nm bandwidth was used as the light source and coupled into an Axiovert 100 microscope (Carl Zeiss, Jena, Germany). The light delivery was controlled by a VS25 and VCM-D1 shutter system (Vincent Associates, Rochester, NY). Light intensities were adjusted manually for light titration experiments using neutral density filters. In recordings of action spectra a motorized filter wheel (Newport, Irvine, CA) was used to ensure delivery of equal photon densities. Light intensities in light titration experiments were measured in the sample plane using a P9710 optometer (Gigahertz Optik, Türkenfeld, Germany) and calculated for the illuminated field ($0.066 \ mm^2$) of the W Plan-Apochromat ×40/1.0 Differential Interference Contrast (DIC) objective (Carl Zeiss).

Measurements were always started in standard buffer (extracellular (in mM): 110 NaCl, 1 KCl, 1 CsCl, 2 CaCl$_2$, 2 MgCl$_2$, 10 2-[4-(2-hydroxyethyl)piperazin-1-yl]ethanesulfonic acid (HEPES); intracellular (in mM): 110 NaCl, 1 KCl, 1 CsCl, 2 CaCl$_2$, 2 MgCl$_2$, 10 HEPES, 10 ethylene-diamine-tetraacetic acid (EDTA)). The pH was adjusted to 7 using *N*-methyl-D-glucamine (NMDG) or citric acid. The osmolarity was measured (Osmomat 3000basic, Gonotec, Berlin, Germany) and adjusted to 320 mOsm (extracellular) or 290 mOSM (intracellular) using glucose. For ion selectivity measurements, NaCl was replaced by 110 mM NaAsp or 110 mM NMDGCl extracellularly. The extracellular buffer was exchanged manually by adding at least 3 ml to the measuring chamber (volume ~0.5 ml). Excess buffer was removed using a Ringer Bath Handler MPCU (Lorenz Messgerätebau, Katlenburg-Lindau, Germany).

Action spectra were recorded in standard conditions at −60 mV using 10 ms light flashes of low intensity (typically in the low µW range at $\lambda_{max}$). $\lambda_{max}$ was determined by normalizing recorded action spectra to the respective maximum photocurrent and applying a Gauss function to the data trimmed between ±70 and ±90 nm from the maximum. To determine ion selectivities, photocurrents were elicited using high-intensity light close to $\lambda_{max}$ applied for 0.5–1 s at −80 to +40 mV increased in steps of 20 mV. The liquid junction potential was calculated and corrected during measurements. Light titrations were recorded at −60 mV with light applied for 1 s. The activity of GtACR1, MsACR1, and raACR with green and red light was measured at −60 mV with optical filters (560 ± 10 nm with 82% transmittance and 692 ± 18 nm with 99% transmittance) inserted directly into the light path.

## Hippocampal neurons cell culture and electrophysiology

Primary cultured hippocampal neurons were prepared, as described (*Karpova et al., 2013*). Neurons were plated (25,000 cells on 12 mm coverslip), maintained at 37°C (5% $CO_2$), and transduced with

crude AAV viruses (1–5 × 10$^{10}$ mg/ml) 3 days after seeding. Experiments were conducted 10–12 days after transduction. Borosilicate glass pipettes (BF100-58-10, Sutter Instruments) were pulled using a micropipette puller (P-2000, Sutter Instruments; 2–3 MΩ). Pipettes were filled using the internal solution as follows (in mM): 135 potassium gluconate, 4 KCl, 2 NaCl, 10 HEPES, 4 ethylene glycol-bis(β-aminoethyl ether)-N,N,N',N'-tetraacetic acid (EGTA), 4 MgATP, 0.3 NaGTP, pH 7.3 (KOH). Tyrode's solution was used as the external solution and prepared as follows (in mM): 125 NaCl, 2 KCl, 30 glucose, 2 MgCl$_2$, 25 HEPES, 2 CaCl$_2$, pH 7.3 (NaOH).

Whole-cell patch-clamp recordings of the ACRs were carried out under visual control with a 12-bit monochrome CMOS camera (Hamamatsu Model OrcaFlash). Current clamp recordings were performed using a MultiClamp700B amplifier, filtered at 8 kHz and digitized at 10 kHz using a Digidata 1550A digitizer (Molecular Devices). Neurons were held at −60 mV, and spiking was evoked with a ramp (600 pA injection; 500 ms) or rectangle protocol (pA injection considering the rheobase +50 pA; 700 ms). The inhibition was tested by activating the ACRs using continuous (700 ms) and pulsed (10 Hz, 2 ms) light and two different wavelengths (550 and 635 nm). The pE-4000 (CoolLED, UK) was used for light stimulation, and the light intensities were measured using a PM100D console and an S130VC sensor (Thorlabs, US).

## Stereotactic injection of viral vectors and optrode implant

Eight-week-old C57BL/6 mice were anesthetized with isoflurane (~1% in O$_2$, vol/vol) and placed into a stereotaxic frame (RWD Life Science). A craniotomy (~1 mm in diameter) anterior-posterior 1.5mm and medial-lateral 1.75mm was made above the injection site. Virus suspensions were slowly injected (100 nl min$^{-1}$) using a 34 G beveled needle (Nanofil syringe, World Precision Instruments). After injection, the needle was left in place for an additional 5 min and then slowly withdrawn. The surgical procedure was continued with the implantation of an optical fiber (Thorlabs 200 um). The optical fiber was secured to the skull using Metabond (Parkell) and dental acrylic. The surgical incision was closed with tissue glue, and 5 mg kg Caprofen was subcutaneously injected for postsurgical analgesia. For AAV/DJ-hSyn1-MsACR-mCerulean and AAV/DJ-hSyn1-raACR-mCerulean, we injected 300 nl with a viral titer of 4.5 × 10$^{12}$ and 1 × 10$^{13}$ vg/ml, respectively.

Self-made optrodes were built according to the design by *Anikeeva et al., 2012*, using an OpenEphys shuttle drive EIB 32-Channel with an Omnetics socket (product #A79026-001) and a 200 µm diameter fiber (Thorlabs CFML). Optrodes were implanted immediately after viral injection, and recordings were started 4 weeks post-implantation.

## Motor cortex behavior

Mice were attached to patch cords before placing them into the middle of the arena (50 cm × 50 cm). Animals were allowed to explore the arena freely for 1 min. Afterward, optical stimulation started with a duration of 2 s at 20 Hz coming from a laser source that was either 473 or 635 nm (MBL-III-473-100 mW and MRL-III-635L-200 mW). We adjusted the light power to 0.5 mW on the 0.03 mm$^2$ tip of the fiber. Stimulation bouts were interspersed with 5-s periods with no light. Stimulation was repeated 25 times before leaving the animal for another 1 min in the arena.

DeepLapCut (*Mathis et al., 2018*) was used to track animals with several body poses (head, ears, snout, body center, and tail). Additionally, we estimated the position of the patch cord, which yielded a drastically improved overall likelihood for body poses. The average pixel deviation between consecutive frames was average for 2 s during stimulation and 2 s after stimulation. Neural recordings were acquired using the OpenEphys system, which was connected to the aforementioned laser. Spike sorting was performed using the Waveclus algorithm (*Chaure et al., 2018*). Z-scores were then calculated for each sorted unit based on 30 trials for each wavelength and light intensity. Units were classified as inhibitory, neutral, or excitatory if the SD of the baseline firing rate differed twofold compared to the first 100 ms between trials.

## Acknowledgements

We thank Sandra Augustin, Celina Dölle, Katharina Draggendorf, Tharsana Tharmalingam, and Maila Reh for outstanding technical assistance, Jonas Wietek for pre-selection of Tara Oceans candidates, and Enrico Peter for help with AlphaFold to generate the MsACR1 structural model,

We are in debt to Ernesto Duran for building and implanting optrodes. This work was supported by the Deutsche Forschungsgemeinschaft (DFG) grant SFB 1078 B2 221545957 (PH) and Israel Science Foundation grant 3131/20 (OB). OB holds the Louis and Lyra Richmond Chair in Life Science. PH is Hertie Senior Professor for Neuroscience supported by the Hertie Foundation. TF was supported by Alexander-von-Humboldt Foundation/CAPES post-doctoral research fellowship (99999.001756/2014-01) and CGC by the EU program 'Novel imaging and brain stimulation methods and technologies' JPco-FuND – NiPARK (MP). MP received additional funding from the Leibniz Association program 'BestMinds' MP J28/2017 SheLi, and Align Science Against Parkinson (ASAP-020505).

## Additional information

### Funding

| Funder | Grant reference number | Author |
|---|---|---|
| Deutsche Forschungsgemeinschaft | SFB 1078 B2 221545957 | Peter Hegemann |
| Israel Science Foundation | 3131/20 | Oded Béjà |
| Alexander von Humboldt-Stiftung | 99999.001756/2014-01 | Thomaz Fabrin |
| EU Joint Programme – Neurodegenerative Disease Research | JPco-FuND - NiPARK | Cristian González-Cabrera Matthias Prigge |
| Leibniz-Gemeinschaft | Program 'BestMinds' MP J28/2017 SheLi | Matthias Prigge |
| Aligning Science Across Parkinson's | ASAP-020505 | Matthias Prigge |

The funders had no role in study design, data collection and interpretation, or the decision to submit the work for publication.

### Author contributions

Johannes Oppermann, Conceived the project, designed molecular characterization, acquired and analyzed electrophysiological data in ND7/23 cells, and wrote the manuscript; Andrey Rozenberg, Conceived the project, performed bioinformatic analyses, and wrote the manuscript; Thomaz Fabrin, Designed neurobiological experiments, acquired and analyzed data in hippocampal neurons in vitro, and contributed to manuscript writing; Cristian González-Cabrera, Designed neurobiological experiments, acquired data from behavioral experiments, and contributed to manuscript writing; Rafael Parker, Performed data analysis of neurobiological experiments and contributed to manuscript writing; Oded Béjà, Conceived the project, acquired funding, and contributed to manuscript writing; Matthias Prigge, Designed neurobiological experiments, performed data analysis, acquired funding, and wrote the manuscript; Peter Hegemann, Conceived the project, designed molecular characterization, and contributed to manuscript writing

### Author ORCIDs

Johannes Oppermann https://orcid.org/0000-0002-3442-1458
Andrey Rozenberg https://orcid.org/0000-0001-9534-2297
Cristian González-Cabrera https://orcid.org/0000-0003-0515-0254
Rafael Parker https://orcid.org/0009-0000-0770-0438
Oded Béjà https://orcid.org/0000-0001-6629-0192
Matthias Prigge https://orcid.org/0000-0002-4923-0056
Peter Hegemann https://orcid.org/0000-0003-3589-6452

### Ethics

All procedures were approved by the Landesverwaltungsamt/Halle and conformed to German regulation. All studies were performed in compliance to animal protocol 42502-2-1545 LIN.

Reviewer #1 (Public review): https://doi.org/10.7554/eLife.90100.3.sa1
Reviewer #2 (Public review): https://doi.org/10.7554/eLife.90100.3.sa2
Author response https://doi.org/10.7554/eLife.90100.3.sa3

## Additional files

### Supplementary files
• MDAR checklist

### Data availability

The bioinformatics workflow is available from the GitHub repository: https://github.com/BejaLab/ACRs (copy archived at *Rozenberg, 2024*). Plasmids of all constructs used in this study were deposited at Addgene: MsACR1-pmCerulean3-N1 (#204958), raACR-pmCerulean3-N1 (#204959), TaraACR1-pmCherry-C1 (#204960), TaraACR2-pmCherry-C1 (#204961), TaraACR3-pmCherry-C1 (#204962), S16-ACR1-pcDNA3.1-mScarlet (#204963), GpACR1-pcDNA3.1-mScarlet (#204964), MsACR1-mCerulean3-pAAV2-hSyn (#204965), raACR-mCerulean3-pAAV2-hSyn (#204966), MsACR1-TS-Kv-P2A-mCerulean3-pAAV-CaMKII (#204970), raACR-TS-Kv-P2A-mCerulean3-pAAV-CaMKII (#204971). The source data of the *in vitro* analysis of channelrhodopsins (Figures 1, 2, Appendix 1—figure 3, and Appendix 1—figure 5) are provided as a source data file. The data and code for analysis of neuro-biological experiments are available from the GitHub repository: https://github.com/TeamPrigge/lab_articles/tree/main/Oppermann_Rozenberg_Fabrin_et_al_24 (https://doi.org/10.5281/zenodo.13905478).

The following dataset was generated:

| Author(s) | Year | Dataset title | Dataset URL | Database and Identifier |
| --- | --- | --- | --- | --- |
| TeamPrigge | 2024 | TeamPrigge/lab_articles: Repository_MsACR_publication_elife_v1.0 | https://doi.org/10.5281/zenodo.13905478 | Zenodo, 10.5281/zenodo.13905478 |

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

## Appendix 1

### Appendix discussion

#### Viral origin of the environmental channelrhodopsins TaraACR1 and TaraACR2

The short metagenomic contigs SAMEA2622336_1814029 (1748 bp) and SAMEA2619952_516956 (1239 bp) encoding the channelrhodopsins (ChRs) TaraACR1 and TaraACR2, respectively, were initially suspected to be of viral origin due to lack of introns and short intergenic regions to the neighboring genes. The neighboring genes, a gene for a DUF285 protein (SAMEA2622336_1814029) and a short fragment of a gene for a Pif1 helicase (SAMEA2619952_516956), are from protein families widespread among, albeit not restricted to, giant viruses. More importantly, the promoter sequence of TaraACR1 contained the classical mimiviral early promoter motif AAAATTGA (*Suhre et al., 2005*) characteristic to *Imitervirales* and some other giant viruses, while that of TaraACR2 contained its minor variation AAAAATGA prevalent for example in the genome of Aureococcus anophagefferens virus (*Imitervirales*) (*Roitman et al., 2023*). Finally, blastn searches against the GVMAGs database (*Schulz et al., 2020*) yielded genomes belonging to the *Imitervirales* containing highly similar nucleotide stretches (best hits to GVMAG-M-3300001348-7, $E$-value $3.3 \times 10^{-80}$, 72.5% nucleotide identity, and GVMAG-M-3300023184-101, $E$-value $8.7 \times 10^{-10}$, 90.2% identity, respectively, both belonging to the *Mesomimiviridae*, see *Aylward et al., 2021*), although, none of them contained the ChR genes themselves. TaraACR1 and TaraACR2 represent a second example of ChR acquisition by giant viruses, alongside prasinophyte ACRs in genomes of viruses closely related to Pyramimonas orientalis virus 01b (*Imitervirales* family IM_12 in *Aylward et al., 2021*) and a putative *Algavirales* virus (*Rozenberg et al., 2020*). The appearance of ion-conducting rhodopsins encoded in genomes of giant algal viruses is not restricted to ChRs, as two unrelated cases of functional rhodopsin cation channels/pumps are known from the *Mesomimiviridae* (viral rhodopsins, see *Bratanov et al., 2019*; *Zabelskii et al., 2020*) and the *Coccolithoviridae* (proton-conducting heliorhodopsins, see *Hososhima et al., 2022*).

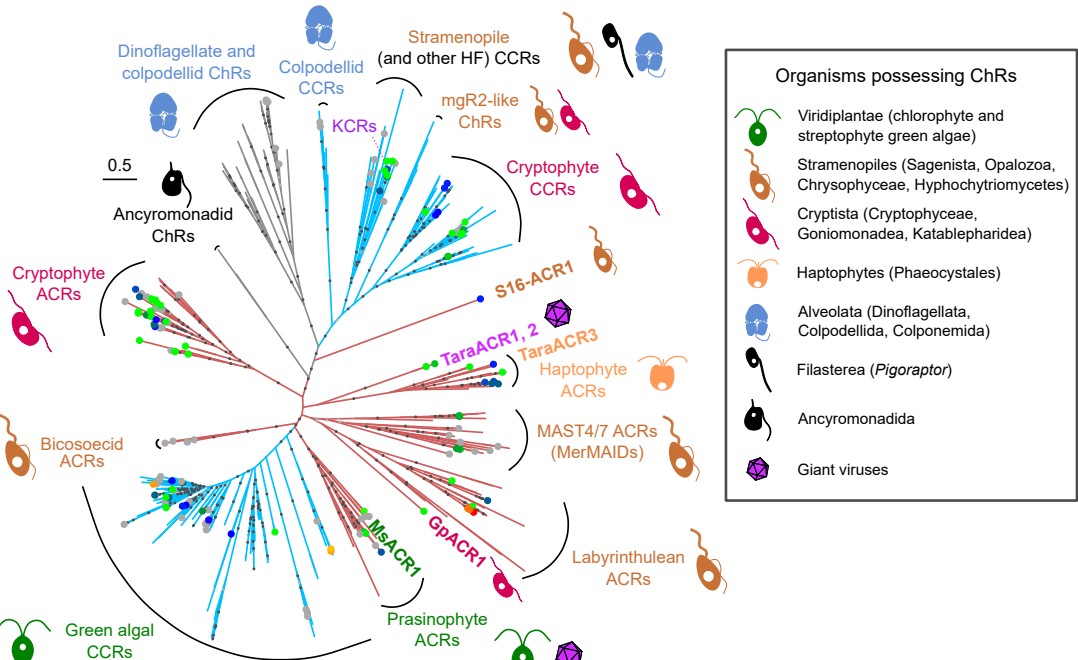

**Appendix 1—figure 1.** Maximum-likelihood phylogenetic reconstruction of relationships among known channelrhodopsins. The sequences chosen for the analysis were taken from the Catalog of Natural Channelrhodopsins (*Rozenberg, 2023*). Names based on the taxonomic affiliation of the organisms possessing the corresponding genes are indicated for major clades. The clades are colored according to the known or putative ion selectivity of their members: red for ACRs and blue for cation-conducting ChRs (CCRs). The color of the dots at the edge tips reflects $\lambda_{max}$ with gray dots corresponding to proteins known to generate photocurrents but having their $\lambda_{max}$ not yet determined. Dots on the internal branches reflect ultrafast bootstrap values: black for ≥95 and gray for ≥90. The positions of the proteins characterized here are highlighted with labels. HF – heterotrophic flagellates.

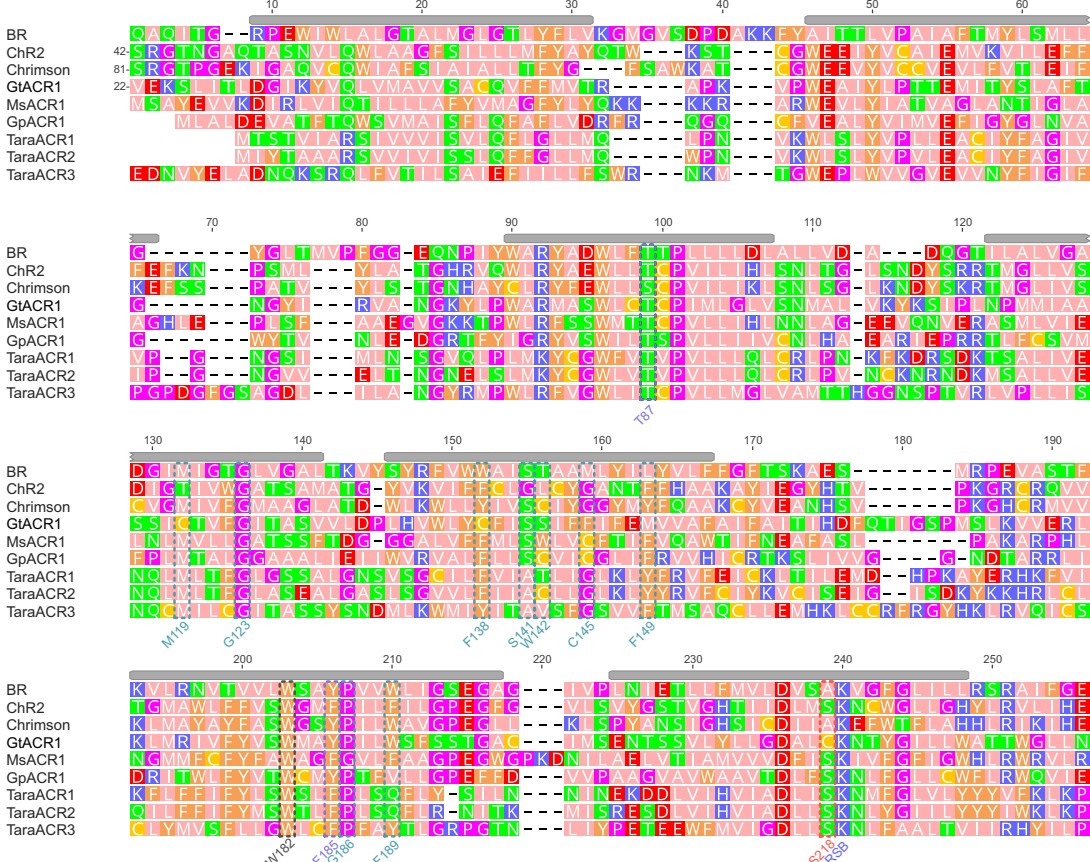

**Appendix 1—figure 2.** Alignment of the rhodopsin domains of the channelrhodopsins (ChRs) tested in the current study and chosen reference proteins. The alignment was constructed using structural information using the crystal structures of BR (PDB: 1AP9), ChR2 (6EID), Chrimson (5ZIH), and GtACR1 (6CSM) and AlphaFold structures for the ChRs studied here. Positions for which MsACR1 mutants were tested are indicated below. The gray bars above the alignment columns indicate transmembrane regions as reported in the OPM database for ChR2.

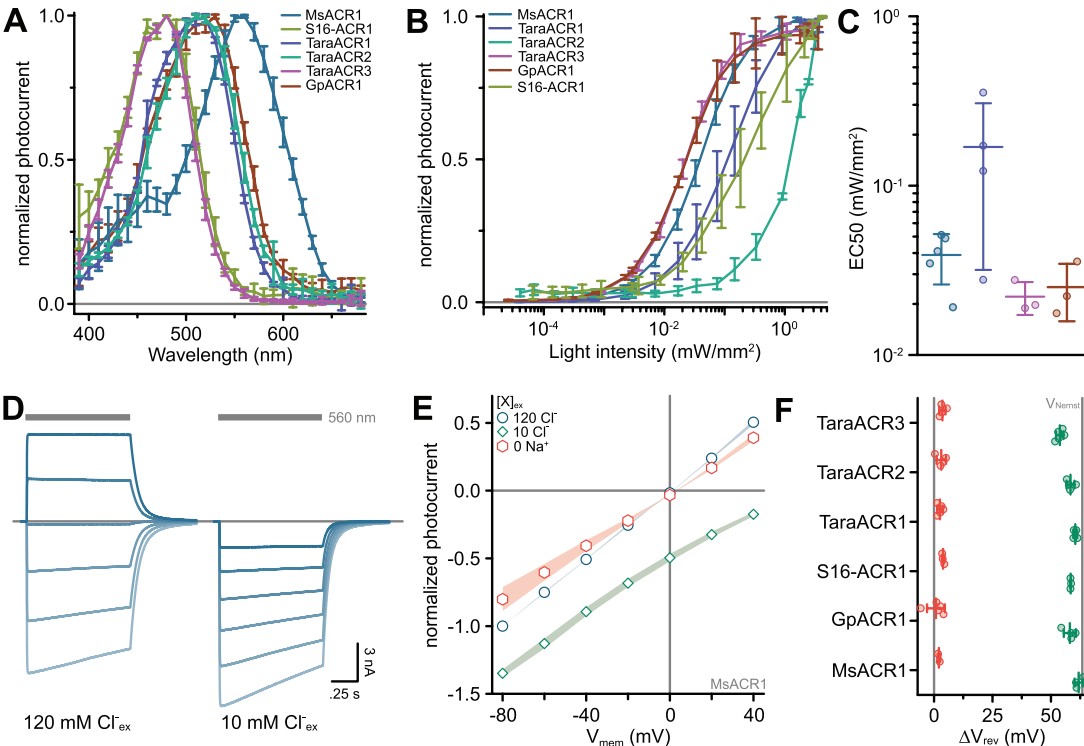

**Appendix 1—figure 3.** Light sensitivity and ion selectivity of identified ACRs. (**A**) Full action spectra of identified ACRs. Shown are mean ± standard deviation connected by lines. (**B**) Light titration of identified ACRs at light intensities between 38 nW/mm² and 3.2 mW/mm² using 1-s light pulses. Shown are mean ± standard error of the mean (SEM) connected by lines. (**C**) Half-maximal activation light intensity determined by the logarithmic fit of light titration curves. Lines are mean ± standard deviation, and circles are data points of single measurements ($n \geq 3$ for all constructs). (**D**) Exemplary photocurrent traces of MsACR1 upon activation with 560 nm light at the indicated extracellular Cl⁻ concentration. (**E**) Current–voltage relationship of MsACR1 at indicated extracellular ion conditions. Shown are mean ± SEM. (**F**) Reversal potential shifts (ΔVrev) upon exchange of the extracellular buffer. Lines are mean ± standard deviation, and circles are data points of single measurements ($n \geq 3$ for all constructs).

The online version of this article includes the following source data for appendix 1—figure 3:

**Appendix 1—figure 3—source data 1.** Source data for electrophysiological analysis to determine light sensitivity and ion selectivity of identified ACRs presented in *Appendix 1—figure 3A-C, E, F*.

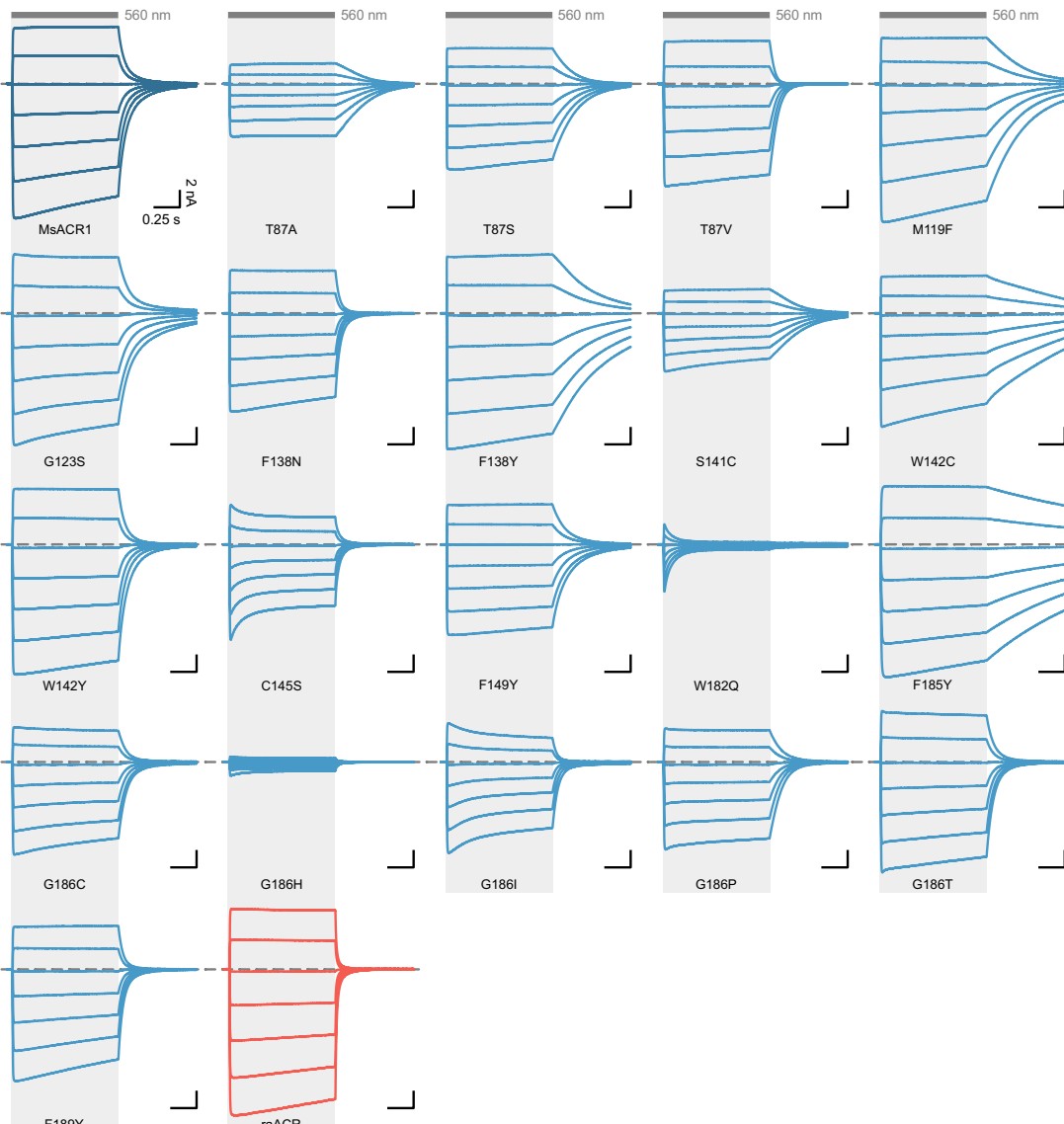

**Appendix 1—figure 4.** Example photocurrent traces of MsACR1 wild-type and mutants. Photocurrent traces of MsACR1 and all mutants tested in this study, measured in ND7/23 cells. Photocurrents were elicited with 560 nm light (gray bars) at membrane voltages between −80 and +40 mV in steps of 20 mV. raACR is MsACR1-S218A.

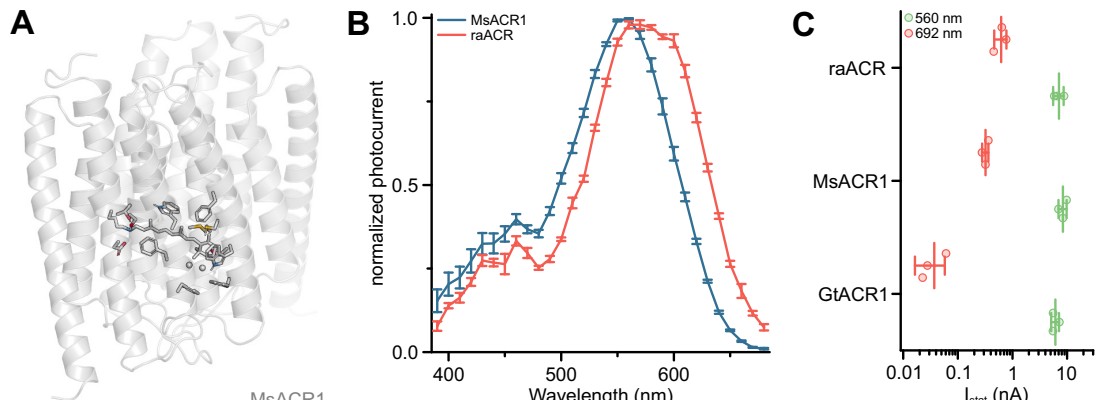

**Appendix 1—figure 5.** The red-light activity of ACRs. (**A**) Structural model of the MsACR1. Residue side chains and the all-*trans*-retinal chromophore are shown as sticks except for G123 and G186, where the Cα is shown as a sphere. Oxygen is colored in red, nitrogen in blue, and sulfur in yellow. (**B**) Full action spectra of MsACR1 and raACR. Shown are mean ± standard error of the mean (SEM) connected by lines. (**C**) Photocurrent amplitude of GtACR1, MsACR1, and raACR upon activation with 560 and 692 nm light. Lines are mean ± standard deviation, and circles are data points of single measurements (*n* = 3 for all constructs).

The online version of this article includes the following source data for appendix 1—figure 5:

**Appendix 1—figure 5—source data 1.** Source data for electrophysiological analysis to determine red-light activity of ACRs presented in *Appendix 1—figure 5B and C*.

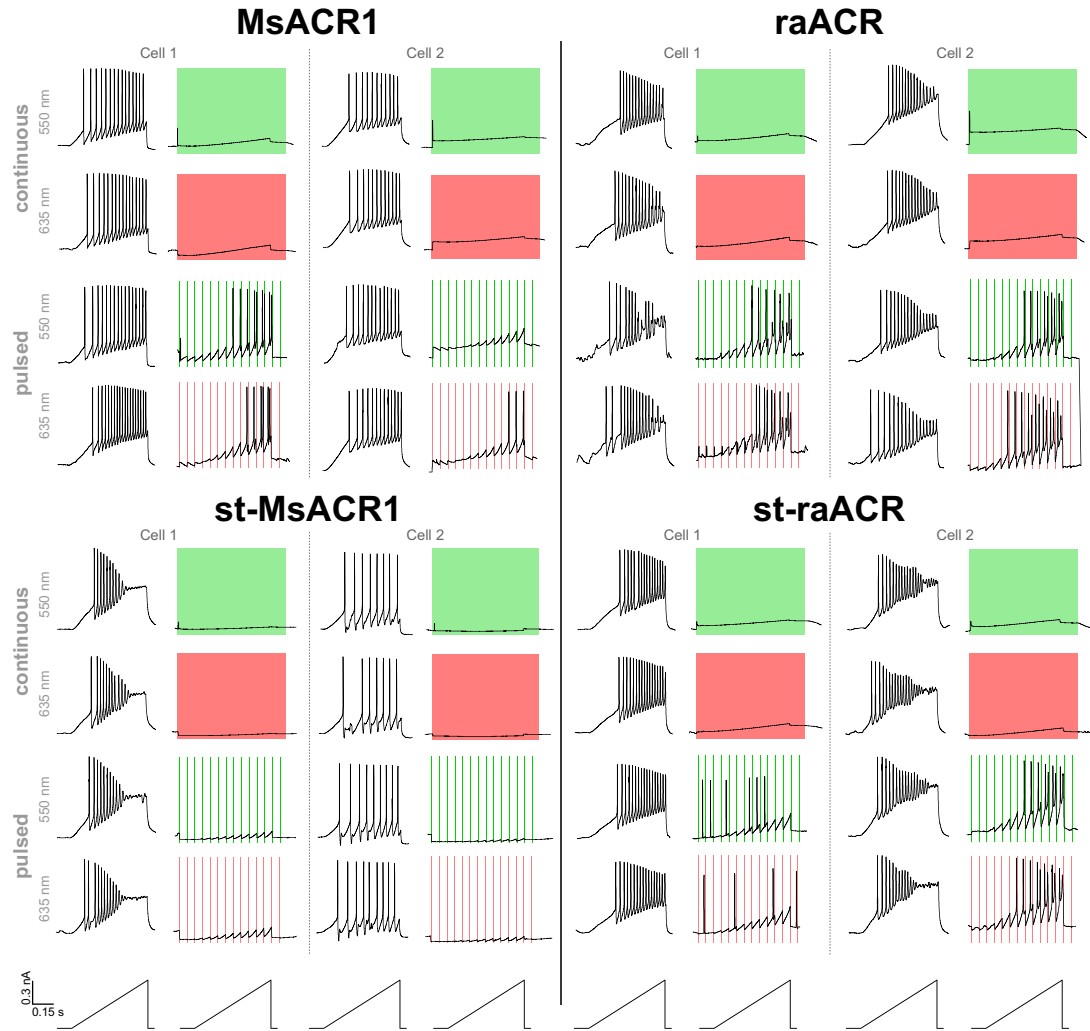

**Appendix 1—figure 6.** Example traces of MsACR1 and raACR in neurons. Voltage traces of MsACR1, raACR, and their soma-targeted versions expressing in hippocampal neurons upon current ramps under no light, continuous or 10 Hz pulsed illumination with 550 or 635 nm light.

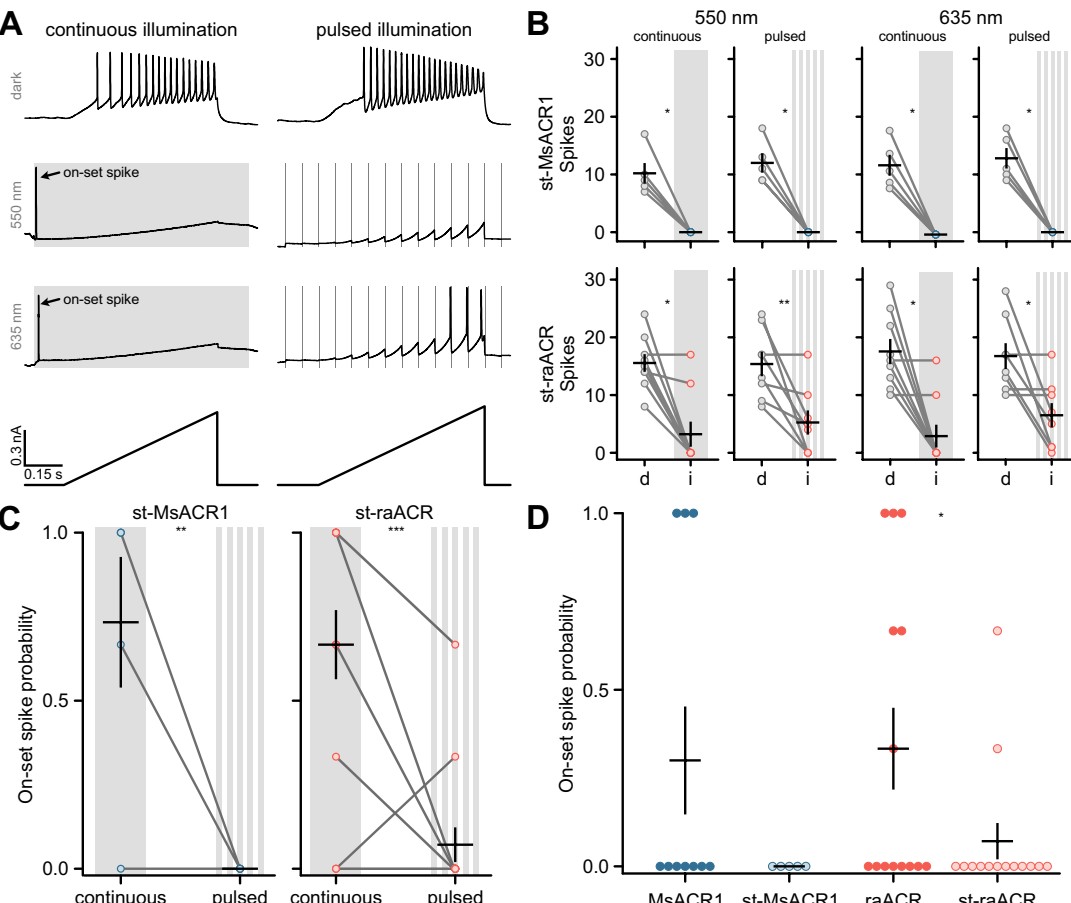

**Appendix 1—figure 7.** Soma targeting of MsACR1 and raACR. (**A**) Stimulation and illumination paradigms used to assess silencing efficiency for soma-targeted MsACR1 and raACR in dissociated hippocampal neurons. (**B**) Spikes elicited by current-ramp injection in no light (**d**) or upon continuous or pulsed illumination (**i**) of 550 or 635 nm in neurons expressing MsACR1 (upper row) or raACR (lower row), as shown in (**A**). Black lines are mean ± standard error, and circles represent single measurements (*n* = 5–9). Paired Wilcoxon or Student's *t*-tests were used for st-MsACR1 and st-raACR comparisons (p < 0.05). (**C**) Wilcoxon-test statistic (one-sided) for the occurrence of a light-evoked spike during continuous and pulsed light stimulation of soma-targeted MsACR1 and raACR (*n* = 5–14), p < 0.05. (**D**) Wilcoxon-test statistic (one-sided) for the occurrence of a light-evoked spike during pulsed light stimulation (*n* = 5–16), ns: p > 0.05, *: p <= 0.05, **: p <= 0.01, ***: p <= 0.001

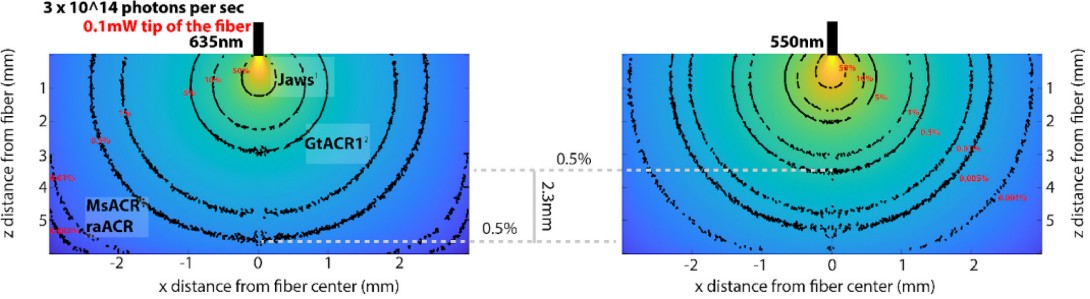

**Appendix 1—figure 8.** Modeling light propagation at 635 and 550 nm via Monte Carlo simulation. To model the propagation of photons with different wavelengths (635 and 550 nm) through brain tissue, we utilized the script developed by *Stujenske et al., 2015*. We initialized the script with a numerical aperture of 0.5 and a fiber radius of 0.1 mm. The absorption coefficient, scatter, and anisotropy parameters were based on the work of *Johansson, 2010*. We simulated 10 million photons at each wavelength (635 nm on the left and 550 nm on the right). The isolines (depicted in black with red labels) indicate the percentage reduction in the number of photons accordingly. *Appendix 1—figure 8 continued on next page*

Comparing this reduction to 0.5% of the initial number of photons leaving the optical fiber reveals that photons at a wavelength of 635 nm travel, on a population level, 2.3 mm further than corresponding photons at 550 nm with similar attenuation. We marked the half-maximum activation light intensities (ED50/mW per mm²) for different inhibitory optogenetic tools: red-shifted chloride pump Jaws (*Chuong et al., 2014*), GtACR1 (*Govorunova et al., 2015*), and MsACR/raACR (from this study). Note: The range of photons is smaller at 635 nm, and the same color space is used for both graphs, making the changes at 635 nm appear to decrease more drastically, which may seem counterintuitive.

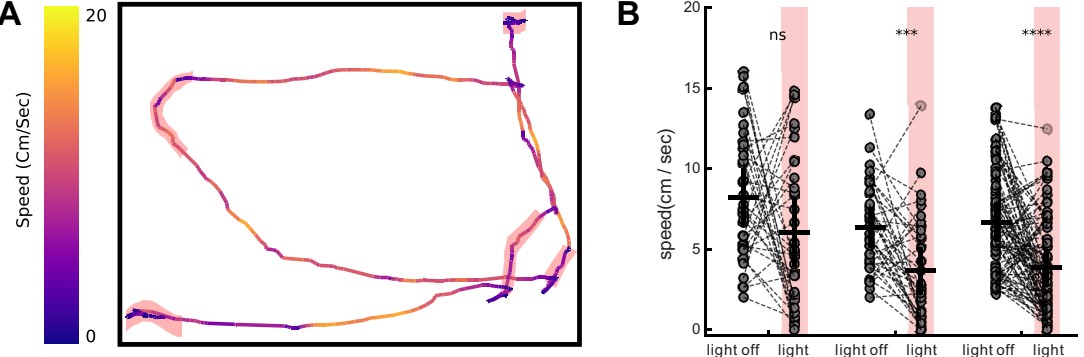

**Appendix 1—figure 9.** Activation of MsACR1 and raACR in PV-positive neurons in primary motor cortex leads to reduced locomotion in OpenField test. (**A**) Speed and trajectory of one example raACR mouse across five stimulation bouts (red highlights). (**B**) Calculated average velocity 2 s prior to and during light stimulation. Each mouse underwent three non-consecutive sessions and only trials in which the mouse had an average speed of 2.5 cm/s prior to stimulation were used to eliminate persistent immobility. Left to right: eGFP control (*n* = 2), MsACR1 (*n* = 2), and raACR (*n* = 3) (Mann–Whitney *U*, ns: p > 0.05, ***: p < 0.001, ****: p < 0.0001).

