## [Editor Report · eLife assessment]

This **important** study describes the discovery and further engineering of a red light-activated, chloride-conducting Channelrhodopsin (ACR) that could be used to inhibit neuronal activity. The evidence for the spectral confirmation and biophysical characterization of MsACR and raACR, and ion selectivity are **solid**; however, the evidence supporting the use of the tools in vivo is **incomplete** and missing proper controls. In addition, benchmarking against other inhibitory tools is somewhat missing. With the in vivo part strengthened, this paper would interest neuroscientists seeking more efficient ways to inhibit neuronal activity.

---

## [Referee Report · Reviewer #1 (Public review)]

Summary:

The authors of this manuscript characterize new anion conducting that is more red-shifted in its spectrum than prior variants called MsACR1. An additional mutant variant of MsACR1 that is renamed raACR has a 20 nm red-shifted spectral response with faster kinetics. Due to the spectral shift of these variants, the authors proposed that it is possible to inhibit expression of MsACR1 and raACR with lights at 635 nm in vivo and in vitro. The authors were able to demonstrate inhibition in vitro and in vivo with 635 nm light. Overall the new variants with unique properties should be able to suppress neuronal activities with red-shifted light stimulation.

Strengths:

The authors were able to identify a new class of anion conducting channelrhodopsin and have variants that respond strongly to lights with wavelength >550 nm. The authors were able to demonstrate this variant, MsACR1, can alter behavior in vivo with 635 nm light.

The second major strength of the study is the development of a red-shifted mutant of MsACR1 that has faster kinetics and 20 nm red-shifted from a single mutation.

Weaknesses:

There are many claims not supported by the evidence provided in the submitted version of the manuscript and would require further experiments to support such claims.

(1) From the data shown, the red-shifted raACR work much less efficiently than MsACR1 even with 635 nm light illumination both in vivo (Figure 4D) and in vitro (Figure 3E) despite the 20 nm red-shift. This is inconsistent with the benefits and effects of red-shifting the spectrum in raACR. The authors claimed that this is due to the faster kinetics of raACR which is plausible from the data shown in Fig 3E but this could be experimentally shown if more examples of continuous illumination and pulsed illumination (such as the one shown in Fig 3D) can be shown in supplemental figures. If this is truly due to the off-kinetics, the spikes would appear after the termination of the pulses but there is little difference in the cases of continuous illumination or during illumination. The fact that 635nm is equally effective as raACR suggests that there is an overall stronger effect of MsACR1 that compensates for the red-shift of raACR.

(2) There are limited comparisons to existing variants of ACRs under the same conditions in the manuscript overall. There should be more parallel comparison with gtACR1, ZipACR and RubyACR in identical conditions in cultured cell line, cultured neurons and in vivo. In terms of overall performance, efficiency, expression in identical conditions. Without this information, it is unclear whether the effects at 635 nm is due to the expression level which can compensate for the spectral shift (which may be the case for MsACR1). The authors stated they are saving this data for another manuscript, this is important data for the current manuscript which should be presented in the existing manuscript.

(3) Despite being able to activate the channelrhodopsin with 635 nm light, the main utility of the variant would be transcranial stimulation which were not demonstrated here.

(4) For the in vivo characterization, there is no mention of animal number and results from Fig 4 and 5 appear to come from multiple samples from a single animal. This is not sufficient scientific evidence to support the claims. Fig 4 and 5 should have statistical analysis from multiple animals and not multiple measurements from single animals in each of the conditions.

(5) As reviewer 2 also pointed out, there is a lack of proper controls (in addition to the low number of animals). The authors point out the current absence of technicians in the laboratory, this should not be a reason to not attempt or do the experiments.

---

## [Referee Report · Reviewer #2 (Public review)]

Summary:

The authors identified a new chloride-conducting Channelrhodopsin (MsACR1) that can be activated at low light intensities and within the red part of the visible spectrum. Additional engineering of MsACR1 yielded a variant (raACR1) with increased current amplitudes, accelerated kinetics, and a 20nm red-shifted peak excitation wavelength. Stimulation of MsACR1 and raACR1 expressing neurons with 635nm in mice's primary motor cortices inhibited the animals' locomotion.

Strengths:

The in vitro characterization of the newly identified ACRs is very detailed and confirms the biophysical properties as described by the authors. Notably, the ACRs are very light sensitive and allow for efficient in vitro inhibition of neurons in the nano Watt/mm^2 range. These new ACRs give neuroscientists and cell biologists a new tool to control chloride flux over biological membranes with high temporal and spatial precision. The red-shifted excitation peaks of these ACRs could allow for multiplexed application with blue-light excited optogenetic tools such as cation-conducting channelrhodopsins or green-fluorescent calcium indicators such as GCaMP.

Weaknesses:

(1) The in-vivo characterization of MsACR1 and raACR1 lacks critical control experiments and is, therefore, too preliminary. The experimental conditions differ fundamentally between in vitro and in vivo characterizations. For example, chloride gradients differ within neurons which can weaken inhibition or even cause excitation at synapses, as pointed out by the authors. Notably, the patch pipettes for the in vitro characterization in neurons contained low chloride concentrations that might not reflect possible conditions found in vivo preparations, i.e., increasing chloride gradients from dendrites to synapses.

2nd review: The authors have addressed this comment in their rebuttal.

(2) Interestingly, the authors used soma-targeted (st) MsACR1 and raACR1 for some of their in vitro characterization yielding more efficient inhibition and reduction of co-incidental "on-set" spiking. Still, the authors do not seem to have utilized st-variants in vivo.

2nd review: The authors offered an explanation in their rebuttal and aim to add these experiments later.

(3) Most importantly, critical in vivo control experiments, such as negative controls like GFP or positive controls like NpHR, are missing. These controls would exclude potential behavioral effects due to experimental artifacts. Moreover, in vivo electrophysiology could have confirmed whether targeted neurons were inhibited under optogenetic stimulations.

Some of these concerns stem from the fact that the pulsed raACR stimulation at 635 nm at 10Hz (Fig. 3E) was far less efficient compared to MsACR1, yet the in vivo comparison yielded very similar results (Fig. 4D).

Also, the cortex is highly heterogeneous and comprises excitatory and inhibitory neurons. Using the synapsin promoter, the viral expression paradigm could target both types and cause differential effects, which has not been investigated further, for example, by immunohistochemistry. An alternative expression system, for example, under VGLUT1 control, could have mitigated some of these concerns.

2nd review: The authors have not added control experiments in Fig 4 yet but plan to conduct them later. They added a new set of experiments in Fig.5 in which PV neurons were exclusively targeted. However, the authors show in vivo electrophysiology demonstrating the expected inhibition of firing neurons (presumably PV neurons expressing raACR) under 635 nm light and disinhibition (presumably excitatory neurons).

(4) Furthermore, the authors applied different light intensities, wavelengths, and stimulation frequencies during the in vitro characterization, causing varying spike inhibition efficiencies. The in vivo characterization is notably lacking this type of control. Thus, it is unclear why the 635nm, 2s at 20Hz every 5s stimulation protocol, which has no equivalent in the in vitro characterization, was chosen.

2nd review: The authors offered a satisfactory explanation.

In summary, the in vivo experiments did not confirm whether the observed inhibition of mouse locomotion occurred due to the inhibition of neurons or experimental artifacts.

2nd review: New experiments were added which demonstrate the expected inhibition of raACR expressing PV neurons in vivo.

In addition, the author's main claim of more efficient neuronal inhibition would require to threshold MsACR1 and raACR1 against alternative methods such as the red-shifted NpHR variant Jaws or other ACRs to give readers meaningful guidance when choosing an inhibitory tool.

The light sensitivity of MsACR1 and raACR1 are impressive and well characterized in vitro. However, the authors only reported the overall light output at the fiber tip for the in vivo experiments: 0.5 mW. Without context, it is difficult to evaluate this value. Calculating the light power density at certain distances from the light fiber or thresholding against alternative tools such as NpHR, Jaws, or other ACRs would allow for a more meaningful evaluation.

2nd review: The authors added Supl Fig. 8 in which light power density and propagation of photons at 550 nm and 635nm through brain tissue was calculated when emitted from light fibers of varying diameter.

---

## [Author Response]

The following is the authors’ response to the original reviews.

**Public Reviews:**

**Reviewer #1 (Public Review):**
Summary:The authors of this manuscript characterize new anion conducting that is more red-shifted in its spectrum than prior variants called MsACR1. An additional mutant variant of MsACR1 that is renamed raACR has a 20 nm red-shifted spectral response with faster kinetics. Due to the spectral shift of these variants, the authors proposed that it is possible to inhibit the expression of MsACR1 and raACR with lights at 635 nm in vivo and in vitro. The authors were able to demonstrate some inhibition in vitro and in vivo with 635 nm light. Overall the new variants with unique properties should be able to suppress neuronal activities with red-shifted light stimulation.Strengths:The authors were able to identify a new class of anion conducting channelrhodopsin and have variants that respond strongly to lights with wavelength >550 nm. The authors were able to demonstrate this variant, MsACR1, can alter behavior in vivo with 635 nm light. The second major strength of the study is the development of a red-shifted mutant of MsACR1 that has faster kinetics and 20 nm red-shifted from a single mutation.Weaknesses:The red-shifted raACR appears to work much less efficiently than MsACR1 even with 635 nm light illumination both in vivo (Figure 4) and in vitro (Figure 3E) despite the 20 nm red-shift. This is inconsistent with the benefits and effects of red-shifting the spectrum in raACR. This usually would suggest raACR either has a lower conductance than MsACR1 or that the membrane/overall expression of raACR is much weaker than MsACR1. Neither of these is measured in the current manuscript.

Thank you for addressing this crucial issue. We posit that the diminished efficiency of raACR in comparison to MsACR1 WT can be attributed to the tenfold acceleration of its photocycle. As noted by Reviewer 1, the anticipated advantages associated with a red-shifted opsin, particularly in in vivo preparations, are offset by its accelerated off-kinetics. Consequently, the shorter dwell time of the open state leads to a reduced number of conducted ions per photon. Nevertheless, the operational light sensitivity is not drastically altered compared to MsACR WT (Fig. 3C). We believe that the rapid kinetics offer interesting applications, such as the precise inhibition of single action potentials through holography.

There are limited comparisons to existing variants of ACRs under the same conditions in the manuscript overall. There should be more parallel comparison with gtACR1, ZipACR, and RubyACR in identical conditions in cultured cell lines, cultured neurons, and in vivo. This should be in terms of overall performance, efficiency, and expression in identical conditions. Without this information, it is unclear whether the effects at 635 nm are due to the expression level which can compensate for the spectral shift.

We compared MsACR1 and raACR with GtACR1 in ND cells in supplemental figure 4. We concur that further comparisons could be useful to emphasise both the strengths of MsACRs and applications where they may not be as suitable. We are currently in the process of outlining a separate article. We firmly believe that each ACR variant occupies a distinct application niche, which necessitates a more comprehensive electrophysiological comparison to provide valuable insights to the scientific community.

There should be more raw traces from the recordings of the different variants in response to short pulse stimulation and long pulse stimulation to different wavelengths. It is difficult to judge what the response would be like when these types of information are missing.

We appreciate Reviewer 1's feedback and have compiled a collection of raw photoresponses, encompassing various pulse widths and wavelengths, which can be found in the Supplementary materials (Supplementary Figures 4 and 5).

Despite being able to activate the channelrhodopsin with 635 nm light, the main utility of the variant should be transcranial stimulation which was not demonstrated here.

We concur with Reviewer 1's assessment that MsACR prime application is indeed transcranial stimulation. However, it's worth emphasising that the full advantages of transcranial optical stimulation become most apparent when animals are truly freely moving without any tethered patch cords. Our ongoing research in the laboratory is dedicated to the development of a wireless LED system that can be securely affixed to the animal's skull. We aim to demonstrate the potential of these novell optogenetic approaches in the field of behavioural neuroscience in the coming year.

Figure 3B is not clearly annotated and is difficult to match the explanation in the figure legend to the figure. The action potential spikings of neurons expressing raACR in this panel are inhibited as strongly as MsACR1.

We have enhanced the figure caption and annotations for clarity. The traces presented in Figure 3B are intended to demonstrate the overall effectiveness of each variant. However, it is in the population data analysis, as depicted in Figure 3E, where the meaningful insights are revealed.

For many characterizations, the number of 'n's are quite low (3-7).

We acknowledge Reviewer 1's suggestion regarding the in vivo data and agree with the importance of including more animals, as well as control animals. However, we are committed to adhering to the principles of the 3Rs (Replacement, Reduction, Refinement) in animal research, and given the robustness of our observed effects, we will add animals to reach the minimal number of animals per condition (n = 2) to minimise unnecessary animal usage while ensuring statistical power.

We will continue to adhere to the established standards in the field, aiming for a range of 3 to 7 cells per condition, sourced from at least two independent preparations, to ensure the robustness and reliability of our in vitro data.

**Reviewer #2 (Public Review):**
Summary:The authors identified a new chloride-conducting Channelrhodopsin (MsACR1) that can be activated at low light intensities and within the red part of the visible spectrum. Additional engineering of MsACR1 yielded a variant (raACR1) with increased current amplitudes, accelerated kinetics, and a 20nm red-shifted peak excitation wavelength. Stimulation of MsACR1 and raACR1 expressing neurons with 635nm in mice's primary motor cortices inhibited the animals' locomotion.Strengths:The in vitro characterization of the newly identified ACRs is very detailed and confirms the biophysical properties as described by the authors. Notably, the ACRs are very light sensitive and allow for efficient in vitro inhibition of neurons in the nano Watt/mm^2 range. These new ACRs give neuroscientists and cell biologists a new tool to control chloride flux over biological membranes with high temporal and spatial precision. The red-shifted excitation peaks of these ACRs could allow for multiplexed application with blue-light excited optogenetic tools such as cation-conducting channelrhodopsins or green-fluorescent calcium indicators such as GCaMP.Weaknesses:The in-vivo characterization of MsACR1 and raACR1 lacks critical control experiments and is, therefore, too preliminary. The experimental conditions differ fundamentally between in vitro and in vivo characterizations. For example, chloride gradients differ within neurons which can weaken inhibition or even cause excitation at synapses, as pointed out by the authors. Notably, the patch pipettes for the in vitro characterization contained low chloride concentrations that might not reflect possible conditions found in the in vivo preparations, i.e., increasing chloride gradients from dendrites to synapses.

We appreciate Reviewer 2’s feedback regarding missing control experiments. We will respond to these concerns in another section of our manuscript, as suggested.

Regarding the chloride gradient, we understand the concerns of Reviewer 2, yet we chose these ionic conditions, particularly as they were used in the initial electrical characterization of GtACR1 in a neuronal context (Mahn et al., 2016). We will make sure to provide this context in our manuscript to justify our choice of ionic conditions.

Interestingly, the authors used soma-targeted (st) MsACR1 and raACR1 for some of their in vitro characterization yielding more efficient inhibition and reduction of co-incidental "on-set" spiking. Still, the authors do not seem to have utilized st-variants in vivo.

At the time of submission, due to the long-term absence of our lab technician, we were not able to produce purified viruses. Therefore, we decided to move on with the submission. We now produced the virus externally, and will provide the experiments.

Most importantly, critical in vivo control experiments, such as negative controls like GFP or positive controls like NpHR, are missing. These controls would exclude potential behavioral effects due to experimental artifacts. Moreover, in vivo electrophysiology could have confirmed whether targeted neurons were inhibited under optogenetic stimulations.

We have several non-injected control animals that we used to calibrate this particular paradigm and never saw similar responses. However, we acknowledge the suggestion of Reviewer 2 and will include the GFP-injected control as recommended.

Some of these concerns stem from the fact that the pulsed raACR stimulation at 635 nm at 10Hz (Fig. 3E) was far less efficient compared to MsACR1, yet the in vivo comparison yielded very similar results (Fig. 4D).

As outlined previously, the accelerated photocycle of raACR results in a reduction in photocurrent amplitude, consequently diminishing the potency of inhibition per photon. In the context of in vitro stimulation, where single action potentials are recorded, this reduction in inhibition efficiency is resolved. However, in the realm of in vivo behavioural analysis, the observed effect is not contingent on single action potentials but rather stems from the disruption of the entire M1 motor network. In this context, despite the reduced efficiency of the fast-cycling raACR, it still manages to interrupt the M1 network, leading to similar behavioural outcomes.

Also, the cortex is highly heterogeneous and comprises excitatory and inhibitory neurons. Using the synapsin promoter, the viral expression paradigm could target both types and cause differential effects, which has not been investigated further, for example, by immunohistochemistry. An alternative expression system, for example, under VGLUT1 control, could have mitigated some of these concerns.

Indeed, we acknowledge the limitations of our current experimental approach. We are in the process of planning and conducting additional experiments involving cre-dependent expression of st-MSACR and st-raACR in PV-Cre mice.

Furthermore, the authors applied different light intensities, wavelengths, and stimulation frequencies during the in vitro characterization, causing varying spike inhibition efficiencies. The in vivo characterization is notably lacking this type of control. Thus, it is unclear why the 635nm, 2s at 20Hz every 5s stimulation protocol, which has no equivalent in the in vitro characterization, was chosen.

We appreciate the valuable comment from the reviewer. The objective of our in vitro characterization is to elucidate the general effects of specific stimulation parameters on the efficiency of neuronal inhibition. For instance, we aim to demonstrate that lower light intensities result in less efficient inhibition, or that pulse stimulation may lead to a less complete inhibition, albeit significantly reducing the energy input into the system.

In the in vivo characterization, we face constraints such as animal welfare considerations and limitations in available laser lines, which prevent us from exploring the entire parameter space as comprehensively as in the in vitro preparation. Additionally, it is important to note that membrane capacitance tends to be higher in vivo compared to dissociated hippocampal neurons. Consequently, we have opted for a doubled stimulation frequency from 10 Hz to 20 Hz and the stimulation pattern of 2 seconds ”on” and 5 seconds “off”. This approach allows the animals to spend less time in an arrested state while still demonstrating the effect of MsACR and variants.

In summary, the in vivo experiments did not confirm whether the observed inhibition of mouse locomotion occurred due to the inhibition of neurons or experimental artifacts.In addition, the author's main claim of more efficient neuronal inhibition would require them to threshold MsACR1 and raACR1 against alternative methods such as the red-shifted NpHR variant Jaws or other ACRs to give readers meaningful guidance when choosing an inhibitory tool.The light sensitivity of MsACR1 and raACR1 are impressive and well characterized in vitro. However, the authors only reported the overall light output at the fiber tip for the in vivo experiments: 0.5 mW. Without context, it is difficult to evaluate this value. Calculating the light power density at certain distances from the light fiber or thresholding against alternative tools such as NpHR, Jaws, or other ACRs would allow for a more meaningful evaluation.

We thank the reviewers for their comments.

**Reviewer #1 (Recommendations For The Authors):**
The study would be much strengthened if the authors can perform more experiments and characterization to support their claims, in addition to showing more raw electrophysiological traces/results and not just summary charts and graphs.

As outlined above, further experiments are planned. We appreciate the suggestion to include more raw electrophysiological traces. Photocurrent traces of all included mutants of MsACR1 measured in ND cells and traces of hippocampal neuronal measurements of non- and soma-targeted MsACR1 and raACR will be included as supplemental figures.

**Reviewer #2 (Recommendations For The Authors):**
Major concern:It is unclear if the optogenetic light stimulation in Fig. 4 caused direct inhibition of neuronal activity in M1, which cell types were targeted, and how MsACR1 and raACR1 compare to other optogenetic inhibitors.Also, the rationale for the light stimulation (635 nm, 2s, 20Hz, every 5s) is not clear.I would suggest the following to address these concerns:(1) M1 expression and stimulation of a negative control such as GFP to exclude that experimental artifacts cause the observed behavioral outcomes.

We are now preparing the required GFP control, and will add it to the new version of the manuscript.

(2) Expression and stimulation of NpHR as a positive control.

We will use st-GtACR1 as a positive control.

(3) Electrophysiological measurements of neuronal activity under optogenetic stimulation to confirm the effectiveness of neuronal inhibition, i.e. suppression of spontaneous firing under light etc.

We concur with Reviewer 2 regarding the potential value of incorporating such in vivo optrode recordings into our manuscript to enable readers to assess the effectiveness of MsACR. As part of our plan for the next version of the manuscript, we intend to conduct these experiments.

(4) ChR2 or other cation-conducting channelrhodopsins with the same expression paradigm could be used to observe diametrically opposite effects.

As Reviewer 2 has already pointed out, the complex interactions that can occur in our viral strategy when an inhibitory opsin is expressed in both excitatory and inhibitory neurons make us sceptical about the possibility of an excitatory opsin leading to opposing effects.

Considering the non-linear input-output function of cortical circuits, optogenetic activation of neurons, even when expressed in either inhibitory or excitatory neurons, is likely to result in the perturbation of the cortical network, which will likely also lead to locomotor arrest.

(5) The authors should confirm whether the expression under synapsin preferentially targeted excitatory and inhibitory cells because inhibiting inhibitory cells could lead to the disinhibition of the principal cells. Synapsin promoters can drive expression in glutamatergic and GABAergic neurons. An alternative expression system under VGLUT1 promoter could yield better targeting.

We concur with Reviewer 2 and will conduct the next set of experiments using the PV-Cre mouse line. Additionally, we will employ in vivo electrophysiology to further confirm the inhibition of the motor cortex network.

(6) Titrating of optogenetic stimulation: The author should test whether increasing or decreasing light intensities and stimulation frequencies as well as different wavelengths (550 nm vs 635 nm) cause differences in inhibiting locomotion in vivo as it did for inhibiting the neuronal firing in vitro (Fig. 3B-E).

The non-linear input-output function within cortical networks, coupled with our sole reliance on behaviour as a readout, will pose challenges in resolving subtle effects on locomotion arrest across various stimulation parameters.

For our planned in vivo electrophysiology recordings, we will measure cortical firing rates as a proxy rather than relying solely on behavioural observations. This approach will allow us to map the fundamental axes of our parameter space in vivo, considering factors such as wavelength, light intensity, and frequency

(7) Explanation of why the 20Hz/2s light stimulation protocol was chosen.

As outlined above, considering animal welfare and increased membrane capacitance in vivo, we opted for the outlined stimulation protocol. This approach allows the animals to spend less time in an arrested state while still demonstrating the effect of MsACR and variants.

(8) In vivo thresholding against other inhibitory tools, such as RubyACRs, Jaws, etc would provide critical guidance for the audience and potential users. It would be particularly important to compare the necessary light intensities for reaching similar behavioral outcomes.

We concur with Reviewer 2 and will prepare data using GtACR1 as a reference.

(9) The author should calculate or reasonably estimate the in vivo light intensity during optogenetic stimulation to provide a meaningful comparison to their in vitro characterization. Ideally, they can provide an estimated volume for efficient stimulation of MsACR1 and raACR1 and compare it to other optogenetic tools.

We will conduct a Monte Carlo simulation and offer a comparison of the effective activation volume across various classes of optogenetic tools.

Minor concerns:(1) Why were st- MsACR1 and raACR1 used in vitro but not in vivo? The viral constructs were described as AAV/DJ-hSyn1-MsACR-mCerulean and AAV/DJ-hSyn1-raACR-mCerulean.

As mentioned earlier, we were unable to produce purified soma-targeted MsACR variants before the manuscript submission. We will now provide these measurements.

(2) Light intensities for the spectral measurements are missing.

During action spectra measurements, a motorised neutral density filter wheel is used to have equal photon flux for all tested wavelengths. Additionally, the light intensity is further reduced by using additional neutral density filters to ensure sufficiently low photocurrents to determine the spectral maximum. Therefore, the light intensity varied between constructs and sometimes measurements. We added the following line to the respective methods section to further clarify this: “(typically in the low µW-range at 𝜆max)”.

(3) MsACR1 is slower and probably more light-sensitive than raACR1, which is faster but has larger photocurrents. These are complementary tradeoffs, and the audience might wonder how MsACR1 and raACR1 photocurrents compare under similar conditions. Therefore, I suggest an alternative representation in Fig. 2C. That is, the presentation of the excitation spectra under similar light intensities and with absolute photocurrent values.

Unfortunately, due to the reasons stated above, MsACR1 and raACR action spectra were not recorded with the same light intensity. However, MsACR1 and raACR are compared under the same conditions for Fig. 2B, E, and F (560 nm light at ~3.2 mW/mm2) as well as in Supp. Fig. 4C.

(4) Figure legends for figures 3F and G are missing details for describing the stimulation paradigm.

We added more details about the stimulation paradigm.